# UNIVAE: A UNIFIED FRAME-ENRICHED VIDEO VAE FOR LATENT VIDEO DIFFUSION MODELS

## ABSTRACT

Variational Autoencoder (VAE) underscores its indispensable role along the growing prominence of Latent Video Diffusion Models (LVDMs). Nevertheless, current latent generative models are generally built upon image VAEs, which compress the spatial dimension only. While, it is vital for video VAE to model temporal dynamic patterns to produce smooth high quality video reconstruction. To address these issues, we propose `UniVAE`, which compresses videos both spatially and temporally while ensuring coherent video construction. Specifically, we employ 3D convolutions at varying scales in the encoder to temporally compress videos, enabling the `UniVAE` to capture dependencies across multiple time scales. Furthermore, existing VAEs only reconstruct videos at a low resolution and fps, bounded by limited GPU memory, which makes the entire video generation pipeline fragmented and complicated. Thus, in conjunction with the new encoder, we explore the potential of the VAE decoder to perform frame interpolation, aiming to synthesize additional intermediate frames without relying on standalone add-on interpolation models. Compared with existing VAEs, the proposed `UniVAE` explores a unified way to compress videos both spatially and temporally with jointly designed encoder and decoder, thus achieving accurate and smooth video reconstruction at a high frame rate. Extensive experiments on commonly used public datasets for video reconstruction and generation demonstrate the superiority of the proposed `UniVAE`. The code and the pre-trained models will be released to facilitate further research.

## 1 INTRODUCTION

The compression and reconstruction of visual data are fundamental to the generative research Rombach et al. (2022); Yu et al. (2023). As latent diffusion models Blattmann et al. (2023); Guo et al. (2023); Yang et al. (2024); Gong et al. (2024) become central to generative tasks, the quality of the latent space largely determines the upper bound of generation performance, which is contingent on the modeling capability of VAE models. While comparing to relatively mature image VAEs Rombach et al. (2022); Podell et al. (2023), video data presents greater challenges due to the larger data volume, temporal redundancy, and GPU memory constraints Chen et al. (2024a). These issues raise the question how to design unified frame-enriched video VAEs dedicated to videos.

Current VAEs can be categorized into two main types: (1) Image VAE approachs Guo et al. (2023); Blattmann et al. (2023), treating a video as a series of individual frames. This kind of VAE effectively preserves spatial pixel contents and typically utilizes lightweight 2D convolution Rombach et al. (2022). However, it may result in temporal instability and inadequate temporal compression in the latent space, as well as large GPU memory consumption of diffusion models. (2) Video VAEs with temporal compression Chen et al. (2024a); Zheng et al. (2024), which can compress video along the temporal dimension, achieving higher compression ratios. Recently, many DiT based video diffusion models Brooks et al. (2024); Lab & etc. (2024) have improved VAEs to achieve better temporal modeling. It greatly increases the number of input tokens, allowing for the generation of videos with more frames. However, the simplistic temporal compression designs in current video VAEs, such as using the primitive temporal pooling operation, limit their ability to effectively capture temporal consistency, considerably restricting the performance of LVDMs. Thus, we believe that a VAE's ability to capture temporal relationships in video data is essential, which shall be designed jointly and integrated in the diffusion generation process for optimal results.

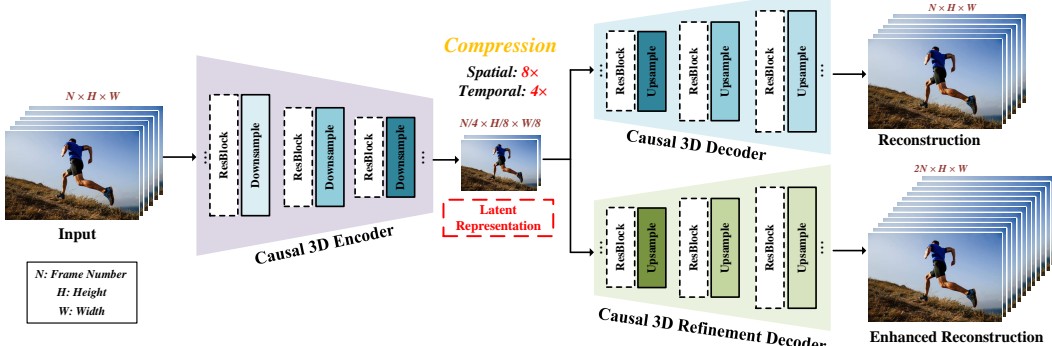

Figure 1: Architecture of the proposed `UniVAE`. It consists of the *Causal 3D Encoder* to compress videos into latent representations for LVDMs temporally and spatially, and the *Causal 3D Decoder* to reconstruct the videos. Moreover, the *Causal 3D Refinement Decoder* is introduced to generate smooth videos, given the low fps outputs by the *Causal 3D Decoder*.

Such a unification is challenging for VAEs, since it involves effectively combining temporal modeling with deep integration into diffusion models. Specifically, the video generation pipeline is much more complicated than that of image generation Brooks et al. (2024); Yang et al. (2024), and is severely bounded by GPU memory. As a result, existing VAEs primarily focus on compressing and reconstructing inputs at a low resolution and frame rate, *e.g.*, 256×256 at 8fps. The interpolation of frames and pixels is delegated to standalone add-on super-resolution Saharia et al. (2022); Shi et al. (2024) or frame interpolation models Reda et al. (2022) connected at the end of the video generation pipeline. This fragmentation complicates the use and optimization of the video generation pipeline, motivating our pursuit of a unified approach. Our observations indicate that VAEs can assume multiple roles within the current video generation stream. If the encoder can effectively model temporal variations, the decoder could theoretically synthesize additional frames, leading to high fps videos without the need for separate interpolation models. This unified VAE structure could reduce the number of frames required by diffusion models to improve training efficiency while streamlining the video generation pipeline.

These issues motivate us to investigate a new approach, `UniVAE`, to compress videos both spatially and temporally and jointly design the encoder and decoder, thus achieving accurate and smooth video reconstruction at a high frame rate. Through the analysis of existing VAEs, we observe that most methods typically utilize single-scale convolution kernels for temporal downsampling Chen et al. (2024a). It limits the ability of VAEs to effectively model motions and changes occurring over varying time scales, making it hard to maintain inter-frame coherence in the reconstructed videos, such as flickering or blurriness. In fact, previous work Yang et al. (2024) has shown that encoding in the temporal dimension in VAEs is more difficult than encoding in the spatial dimension, which suggests that improving the temporal encoding capability is the key challenge.

To this end, we propose a multi-scale temporal convolution architecture for the encoder in `UniVAE`. Specifically, during the temporal compression, we apply 3D convolution Ji et al. (2012) with varying kernel sizes to downsample the videos in the temporal dimension, followed by concatenating and fusing the extracted features. Our multi-scale convolutions are designed across the channel dimension, without introducing additional computational cost. It allows `UniVAE` to capture dynamic patterns across different time scales, enabling a comprehensive understanding of the video's temporal structure. Furthermore, we integrate the frame interpolation function into the decoder to generate high frame rate video, enabling the production of smooth video content. For this purpose, we propose a latent-guided refinement training scheme, which introduces a refinement decoder to `UniVAE` with the assistance of a pre-trained VAE decoder.

We summarize our contributions as follows:

- We propose the `UniVAE` which is designed specifically for video data. Both the encoder and decoder in `UniVAE` are enhanced to compress and reconstruct videos temporally and spatially, devoted to video generation via latent diffusion models.

- Our `UniVAE`'s encoder employs a multi-scale temporal convolution architecture to capture dynamic video representations, while the decoder reconstruct and interpolate video frames simultaneously for smooth video content. To the best of our knowledge, this is the first effort to unify the spatial and temporal modeling in the encoder, and unify reconstruction and interpolation in the decoder in VAE.

- Extensive experiments on commonly used public datasets for video reconstruction and generation demonstrate the superiority of the proposed `UniVAE`.

## 2 RELATED WORKS

**Video Generation Models.** Video generation may find killer applications in short video apps, video ads, and filming. Ho et al. (2020). With the emergence of OpenAI SORA Brooks et al. (2024), the immense potential of video generation models has become evident, prompting researchers to invest passion and resources into this field. Some works Ho et al. (2022); Singer et al. (2022); Zhang et al. (2023) directly generate videos in pixel space. Due to the computational constraints, recent works have shifted toward learning video distributions in a latent space. The popular approach involves leveraging a Variational Autoencoder (VAE) to compress videos into a latent space, followed by employing diffusion models to learn the distribution within that space. The works such as Animate-Diff Guo et al. (2023), Stable Video Diffusion Blattmann et al. (2023) adopt the U-Net structure from T2I diffusion models, while others, such as SORA Brooks et al. (2024), Open-Sora Zheng et al. (2024), Latte Ma et al. (2024), CogVideoX Yang et al. (2024), and Open-Sora-Plan Lab & etc. (2024), utilize transformer as the denoisers. Regardless of the denoiser architecture, VAE plays a decisive role in determining the quality of the latent space for training and the final video reconstruction, which is crucial for latent video diffusion models (LVDMs).

**Variational Autoencoder.** Variational Autoencoder (VAE) Kingma (2013) is designed to generate new data by sampling from learned latent distribution. Recently, it has been widely used in generative models, which can be divided into two categories, discrete and continuous. The first one is discrete VQ-VAEs Van Den Oord et al. (2017), which compress an input sample into a latent space and quantize it into discrete tokens with a codebook. These tokens are then fed into subsequent autoregressive-based generative models Yu et al. (2023). The second one is continuous VAEs, which compress an input sample into a continuous latent space and have been widely used in latent diffusion models. Stable Diffusion Rombach et al. (2022) is the first work to introduce VAE to diffusion models. For LVDMs, a straightforward approach is to apply Stable Diffusion VAE Rombach et al. (2022) to compress input videos frames by frames. Stable Video Diffusion VAE Blattmann et al. (2023) further explore the temporal relations based on Stable Diffusion VAE. However, these VAEs only compress videos spatially, while ignoring the temporal dimension. Recently, several works have explored designing VAEs that compress videos both temporally and spatially. OS-VAE Zheng et al. (2024) adopts a cascaded architecture, where a temporal VAE is applied to the latent space of a spatial VAE. CV-VAE Zhao et al. (2024) proposes to compress videos temporally while focusing on aligning its latent space with that of the existing Stable Diffusion VAE. OD-VAE Chen et al. (2024a) also proposes to compress video temporally. Building on previous research, the proposed `UniVAE` explores a unified way to compress videos both spatially and temporally with jointly designed encoder and decoder, thus achieving accurate and smooth video reconstruction at a high frame rate.

## 3 METHOD

In this section, we will introduce the proposed `UniVAE`. We first overview the whole pipeline of our `UniVAE` in Sec. 3.1. Then, we introduce the multi-scale temporal downsampling in Sec. 3.2 and latent-guided refinement training strategy in Sec. 3.3, respectively.

### 3.1 OVERVIEW

The architecture of the proposed `UniVAE` is illustrated in Fig. 1, which consists of an encoder $\mathcal{E}$ and two decoders $\mathcal{D}_1$ and $\mathcal{D}_2$. We denote a video with $(N+1)$ frames as $\mathbf{X} \in \mathbb{R}^{(N+1) \times H \times W \times 3}$, where $H$ and $W$ are the height and width of each frame $\mathbf{x}_i$ in $\mathbf{X}$. Then, we feed $\mathbf{X}$ into $\mathcal{E}$ to compress it to the latent representation $\mathbf{Z} \in \mathcal{R}^{(n+1) \times h \times w \times c}$, which can be formulated as $\mathbf{Z} = \mathcal{E}(\mathbf{X})$. Following the

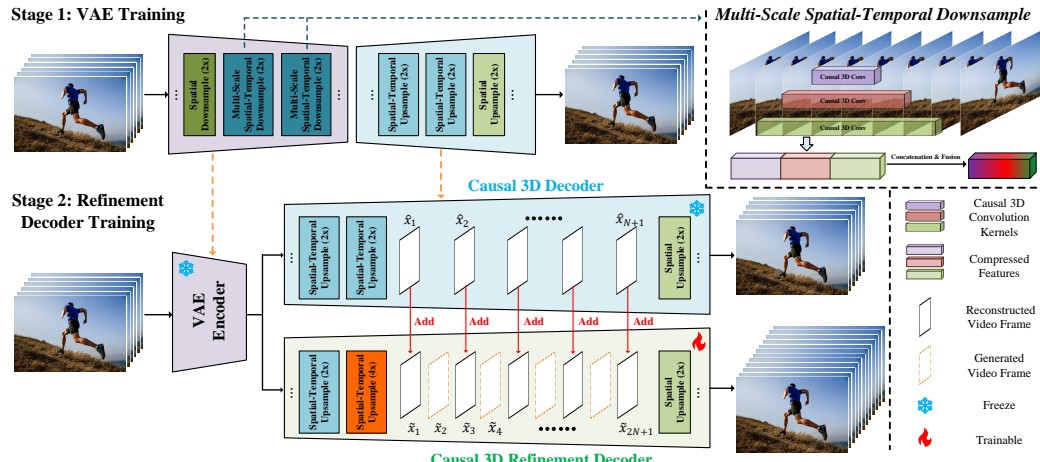

Figure 2: Illustration of the latent-guided refinement training strategy, which is implemented as a 2-stage training. In stage 1, we train the `UniVAE` encoder $\mathcal{E}$ and decoder $\mathcal{D}_1$ for accurate video reconstruction. In stage 2, we freeze the $\mathcal{E}$ and $\mathcal{D}_1$, and train the refinement decoder $\mathcal{D}_2$ independently. The features in $\mathcal{D}_1$ are injected into $\mathcal{D}_2$ to facilitate the frames interpolation.

common setting in 3D VAEs for LVDMs Chen et al. (2024a); Zhao et al. (2024); Zheng et al. (2024), the temporal rate $\rho_t = \frac{N}{n}$ is set to 4, and the spatial rate $\rho_s = \frac{H}{h} = \frac{W}{w}$ is set to 8, respectively. The first frame $\mathbf{x}_1$ will is processed independently, which is only compressed spatially. A multi-scale temporal downsampling is introduced to the causal 3D encoder $\mathcal{E}$ to compress videos temporally and spatially, which will be detailed in Sec. 3.2. Among the two decoders, the causal 3D decoder $\mathcal{D}_1$ is responsible for reconstructing the input videos, while the causal 3D refinement decoder $\mathcal{D}_2$ interpolates to further enhance the smoothness of the reconstructed videos. This can be formulated as $\hat{\mathbf{X}} = \mathcal{D}_1(\mathbf{Z})$ and $\tilde{\mathbf{X}} = \mathcal{D}_2(\mathbf{Z})$, where $\hat{\mathbf{X}} \in \mathbb{R}^{(N+1) \times H \times W \times 3}$ denotes the reconstructed video, and $\tilde{\mathbf{X}} \in \mathbb{R}^{(2N+1) \times H \times W \times 3}$ means the enhanced videos with the interpolation of frames, respectively. The training strategy of the two decoders will be described in Sec. 3.3.

## 3.2 MULTI-SCALE TEMPORAL DOWNSAMPLING

In addition to spatial compression, video VAEs also compress vidoes temporally to further reduce the computational cost of video generation in LVDMs. However, the temporal compression inevitably leads to details loss, resulting in issues such as jitter and flickering in the reconstructed videos. Upon reviewing the existing video VAE architectures, we observe that most of them rely on fixed-size convolutional kernels for temporal downsampling, with a kernel size of 3 being the common choice Chen et al. (2024a). This limits their ability to capture dynamic temporal features across different time scales in videos. In fact, previous work Yang et al. (2024) has shown that temporal compression in video VAEs is more challenging than spatial compression for videos. Therefore, it is crucial to explore better schemes for temporal compression.

To solve this issue, we propose a multi-scale temporal downsampling architecture to compress videos temporally in the causal 3D encoder. Unlike previous works Chen et al. (2024a); Zhao et al. (2024) that only utilize fixed-size convolutional kernels to compress videos temporally, we design a series of convolutional kernels with varying kernel sizes for temporal compression, as shown in Fig. 2. Specifically, we apply multiple causal 3D convolution $\mathbf{F} = \{f_1, f_2, ..., f_p\}$, where $f_i \in \mathbb{R}^{c_{in} \times c_{out} \times t_i \times h_i \times w_i}$ ($i \in [1, ..., p]$). $t_i$, $h_i$, and $w_i$ represent the temporal size, height, and width of the convolution kernel. Each $f_i$ has a unique kernel size $t_i$ along the time axis, which endows $\mathbf{F}$ with the ability to capture dynamic patterns of different time scales in videos. On the other hand, for all $f_i \in \mathbf{F}$, they share the same $h_i$ and $w_i$.

Given an input video tensor $\mathbf{x} \in \mathbb{R}^{N \times H \times W \times C}$, we first partition it into $p$ parts along the channel dimension. We can get $\mathbf{x} = [x_1, x_2, ..., x_p]$, where $x_i \in \mathbb{R}^{N \times H \times W \times \frac{C}{p}}$. Then, we leverage the multi-scale convolution $\mathbf{F}$ to compress $\mathbf{x}$ temporally and spatially. Specifically, we first perform

convolution between each kernel $f_i$ and its corresponding segment $x_i$, and then concatenate the resulting $y_i$ along the channel dimension to produce the final output $\mathbf{y}$, which can be formulated as:

$$\mathbf{y} = \mathbf{F} \otimes \mathbf{x} = [f_1 \otimes x_1, f_2 \otimes x_2, ..., f_p \otimes x_p],$$

where $\otimes$ is the convolution operation. Compared with previous video VAEs, `UniVAE` leverages convolutional kernels with varying scales for video temporal compression. This allows the `UniVAE` to capture temporal dependencies across different time scales. $f_i$ with smaller $t_i$ will focus on local temporal variations, capturing fast movements and transient changes, while those with larger $t_i$ capture long-term dynamics such as slow motions or extended trends. This enhances the diversity and richness of the features extracted by $\mathcal{E}$, providing more varied and informative latent representations for subsequent video diffusion models. Moreover, the multi-scaled features supports the $\mathcal{D}_2$ to interpolate additional frames in between with consistent object appearance and coherent motion, maintaining the coherence of the enhanced videos, which will be discussed in Sec. 3.3. Our design also improves the VAE's generalization ability, enabling it to adapt to videos with different motion speeds and temporal patterns.

### 3.3 LATENT-GUIDED REFINEMENT TRAINING STRATEGY

Existing video VAEs typically employ an encoder to compress a video $\mathbf{X}$ into lower-dimensional latent representation $\mathbf{Z}$ temporally and spatially to reduce the computational cost of LVDMs, and then utilize a decoder to reconstruct the corresponding a video $\hat{\mathbf{X}}$ to ensure high-quality video generation, where $\mathbf{X}$ and $\hat{\mathbf{X}}$ have the same number of frames. **Beyond the common paradigm, we wonder whether 3D VAEs have the potential to generate richer video content beyond reconstruction.** Specifically, by interpolating frames during the decoding process, the decoder can produce content-rich videos $\tilde{\mathbf{X}}$. This could allow LVDMs to generate smooth, high-fps videos without increasing computational costs of diffusion models.

To this end, we introduce a causal 3D refinement decoder to the `UniVAE`, which can interpolate video frames, as illustrated in Fig. 2. In order to train the proposed decoder and refinement decoder effectively, we propose a two-stage latent-guided refinement training scheme. As shown in Fig. 2, we train the VAE encoder $\mathcal{E}$ and decoder $\mathcal{D}_1$ in the first stage. Given the original video $\mathbf{X}$, we send it to the encoder to get the latent representation $\mathbf{Z}$, which is $\mathbf{Z} = \mathcal{E}(\mathbf{X})$. Then, we utilize the decoder to reconstruct video $\hat{\mathbf{X}}$, which is $\hat{\mathbf{X}} = \mathcal{D}_1(\mathbf{Z})$. Following the training of video VAEs in LVDMs Chen et al. (2024a); Zhao et al. (2024), we use a combination of reconstruction loss Zhang et al. (2018), adversarial loss Goodfellow et al. (2020), and KL regularization Kingma (2013) as the training objective, formulated as:

$$\mathcal{L}_{VAE}(\mathcal{E}, \mathcal{D}_1) = \mathcal{L}_{recon}(\mathbf{X}, \hat{\mathbf{X}}) + \mathcal{L}_{adv}(\mathbf{X}, \hat{\mathbf{X}}) + \mathcal{L}_{KL}(\mathbf{X}, \mathbf{Z}), \tag{1}$$

After training $\mathcal{E}$ and $\mathcal{D}_1$, we introduce a refinement decoder $\mathcal{D}_2$ to `UniVAE` and train it independently in the second stage. $\mathcal{E}$ and $\mathcal{D}_1$ are frozen in this stage. The refinement decoder $\mathcal{D}_2$ shares almost the same architecture as the standard decoder $\mathcal{D}_1$, with the primary difference being that $\mathcal{D}_2$ employs a higher temporal upsampling rate in the final temporal decoding module, as illustrated in Fig. 2. This enables $\mathcal{D}_2$ interpolate an extra intermediate frame between every two consecutive reconstructed frames frames, improving the smoothness of the reconstructed videos. Since $\mathcal{D}_1$ has been trained to decode video from $\mathbf{Z}$, we leverage it to facilitate the training of $\mathcal{D}_2$. Specifically, given a video $\mathbf{X} \in \mathbb{R}^{(N+1) \times H \times W \times 3}$, we send it to $\mathcal{E}$ to get $\mathbf{Z}$. Then, we feed $\mathbf{Z}$ into $\mathcal{D}_1$ and $\mathcal{D}_2$, respectively. After passing the last temporal upsampling module, we can get the $\hat{\mathbf{x}} = [\hat{x}_1, \hat{x}_2, ..., \hat{x}_{N+1}]$ in $\mathcal{D}_1$, and $\tilde{\mathbf{x}} = [\tilde{x}_1, \tilde{x}_2, ..., \tilde{x}_{2N+1}]$ in $\mathcal{D}_2$, respectively. For each $\hat{x}_i \in \hat{\mathbf{x}}$, we can find the corresponding frame $\tilde{x}_{2i-1} \in \tilde{\mathbf{x}}$. Then, we add each frame in $\hat{\mathbf{x}}$ to the corresponding frame in $\tilde{\mathbf{x}}$, formulated as:

$$\tilde{\mathbf{x}}_{new} = \tilde{\mathbf{x}} + \hat{\mathbf{x}} = [\tilde{x}_1 + \hat{x}_1, \tilde{x}_2, \tilde{x}_3 + \hat{x}_2, \tilde{x}_4, ..., \tilde{x}_{2i-1} + \hat{x}_i, \tilde{x}_{2i}, ..., \tilde{x}_{2N}, \tilde{x}_{2N+1} + \hat{x}_{N+1}], \tag{2}$$

Then, $\tilde{\mathbf{x}}_{new}$ is fed to subsequent module for further reconstruction. Since $\hat{\mathbf{x}}$ is necessary for reconstructing $\mathbf{X}$, we inject $\hat{\mathbf{x}}$ into $\tilde{\mathbf{x}}$. This makes it easy for $\mathcal{D}_2$ to reconstruct existing frames (*i.e.*, $[\tilde{x}_1, \tilde{x}_3, ..., \tilde{x}_{2N+1}]$), and encourage $\mathcal{D}_2$ to focus on the generation of additional frames (*i.e.*, $[\tilde{x}_2, \tilde{x}_4, ..., \tilde{x}_{2N}]$). The multi-scale latent representation $\mathbf{Z}$ extracted by $\mathcal{E}$ provide the refinement decoder $\mathcal{D}_2$ with details about temporal patterns at various time scales, which helps $\mathcal{D}_2$ make informed additional frame generation about what should occur between existing frames when upsampling. Unlike frame interpolation methods Reda et al. (2022) with a stand-alone interpolation

Table 1: Performance comparison of different VAEs on 25-frame video reconstruction across WebVid-10M and Panda-70M dataset. The best results are marked as **bold**, and the second ones are marked by underline. Note that the **UniVAE** * denotes that the UniVAE w/o refinement decoder $\mathcal{D}_2$, since only $\mathcal{D}_1$ are utilized for standard video reconstruction.

| Method | VCR | Params | WebVid-10M | | | Panda-70M | | |
|---|---|---|---|---|---|---|---|---|
| | | | PSNR↑ | SSIM↑ | LPIPS↓ | PSNR↑ | SSIM↑ | LPIPS↓ |
| VQGAN | $1\times8\times8$ | 69.0M | 26.26 | 0.7699 | 0.0906 | 26.07 | 0.8295 | 0.0722 |
| SD-VAE | $1\times8\times8$ | 83.7M | 30.19 | 0.8379 | 0.0568 | 30.40 | 0.8894 | 0.0396 |
| SVD-VAE | $1\times8\times8$ | 97.7M | 31.15 | 0.8686 | 0.0547 | 31.00 | 0.9058 | 0.0379 |
| TATS | $4\times8\times8$ | 52.2M | 23.10 | 0.6758 | 0.2645 | 21.77 | 0.6680 | 0.2858 |
| CV-VAE | $4\times8\times8$ | 182.5M | 30.76 | 0.8566 | 0.0803 | 29.57 | 0.8795 | 0.0673 |
| OS-VAE | $4\times8\times8$ | 393.3M | 31.12 | 0.8569 | 0.1003 | 31.06 | 0.8969 | 0.0666 |
| OD-VAE | $4\times8\times8$ | 239.2M | 31.16 | 0.8694 | 0.0586 | 30.49 | 0.8970 | 0.0454 |
| **UniVAE** * | $4\times8\times8$ | 234.8M | **34.13** | **0.8783** | **0.0525** | **33.58** | **0.9138** | **0.0444** |

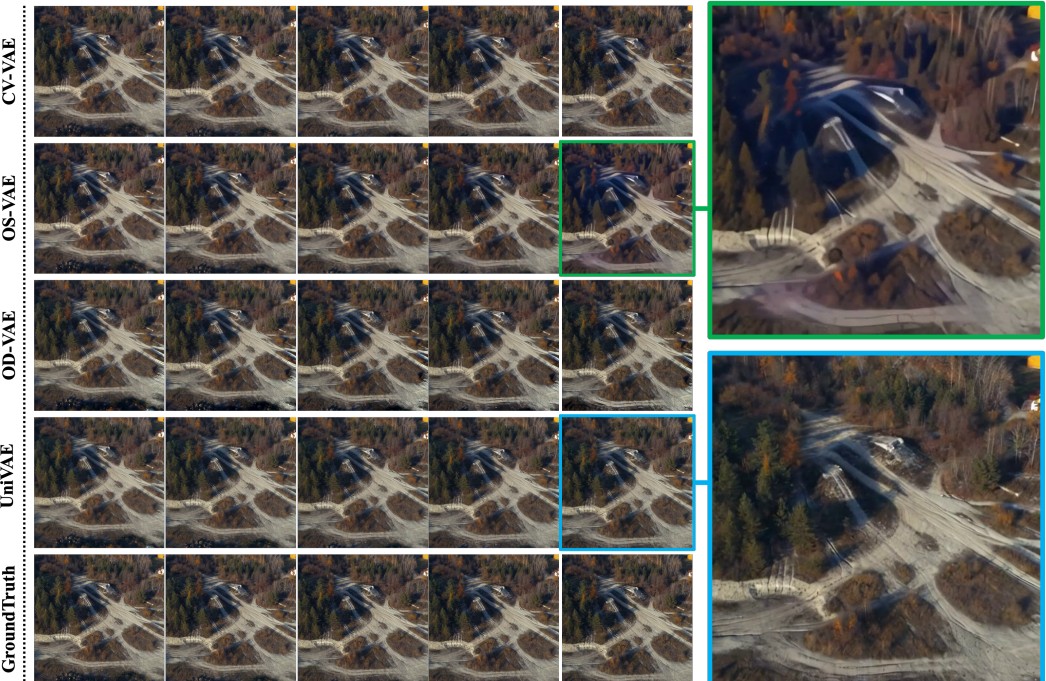

Figure 3: Qualitative comparison of different 3D VAEs on video reconstruction.

module, we integrate the frame interpolation into the VAE decoding process, rather than applying post-processing frame interpolation to the decoded videos. While various intermediate frame methods are available, UniVAE utilizes a straightforward variant that directly upsamples features temporally in $\mathcal{D}_2$ to explore the potential of video VAE to reconstruct videos with high fps.

## 4 EXPERIMENTS

### 4.1 EXPERIMENTAL SETTINGS

**Training Details.** In the first stage, we train the encoder $\mathcal{E}$ and the decoder $\mathcal{D}_1$ of the UniVAE with Adam optimizer for 380k steps, where $\beta_1$ and $\beta_2$ are set to 0.9 and 0.999, respectively. The learning rate is set to a constant value of $1\times10^{-5}$, and the batch size is set to 8. Following Chen et al. (2024a), the input videos are pre-processed to a length of 25 frames with a resolution of $256 \times 256$. In the second stage, we independently train the refinement decoder $\mathcal{D}_2$ for 1,000k steps. In this stage, the input videos are processed to a length 25 frames with a resolution of $128 \times 128$. The entire UniVAE is trained on $8 \times$ NVIDIA A100 GPUs under Pytorch framework.

Table 2: Performance comparison of different VAEs on video generation across UCF101 and Sky-Timelapse dataset. The best results are marked as **bold** and the seconds one are marked by underline.

| Method | UCF101 | | SkyTimelapse | |
|---|---|---|---|---|
| | FVD↓ | KVD↓ | FVD↓ | KVD↓ |
| Latte + CV-VAE | 8742.42 | 20.13 | 986.30 | 11.25 |
| Latte + OD-VAE | 8047.60 | 20.65 | 881.66 | **9.41** |
| Latte + UniVAE | **7777.71** | **19.35** | **799.64** | 9.56 |

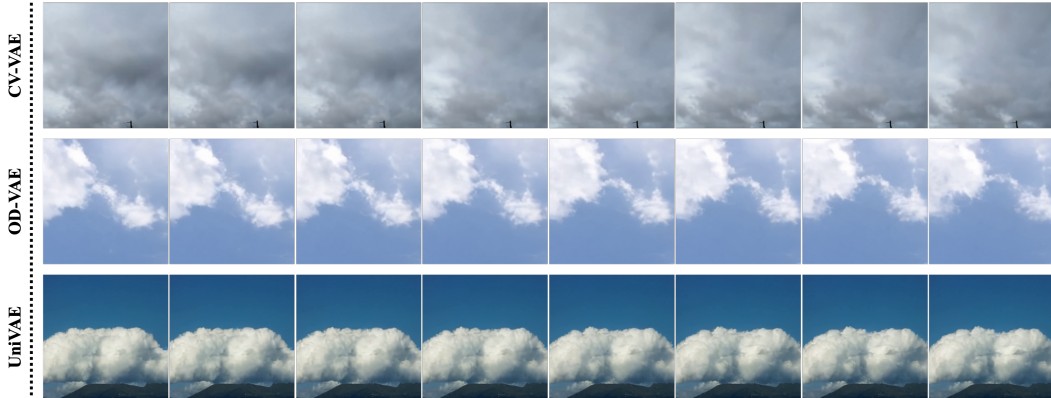

Figure 4: Video generation results of Latte equipped with different VAEs on SkyTimelapse dataset.

**Datasets and Evaluation Metrics.** We evaluate our UniVAE on video reconstruction and video generation. For video reconstruction, WebVid-10M Bain et al. (2021) and Panda-70M Chen et al. (2024b) are utilized for evalution. To assess the performance for video reconstruction, we adopt three different metrics, including PSNR Hore & Ziou (2010), SSIM Wang et al. (2004) and LPIPS Zhang et al. (2018). Among them, PSNR and SSIM are utilized to quantify the fidelity of the reconstructed videos, where higher values mean better reconstruction quality. LPIPS is utilized to measure the perceptual difference and visual quality, with lower values indicating better performance. For video generation, we use the Latte Ma et al. (2024) as the latent video diffusion model. We apply different VAEs to Latte and evaluate their video generation performance. Following Chen et al. (2024a); Ge et al. (2022), the UCF101 Soomro (2012) and SkyTimelapse Xiong et al. (2018) are selected as datasets for video generation evaluation. Frechet Video Distance (FVD) and Kernel Video Distance (KVD) Unterthiner et al. (2018) are employed as the evaluation metrics to assess the performance of video generation.

## 4.2 COMPARISON ON VIDEO RECONSTRUCTION AND VIDEO GENERATION

Tab. 1 shows the performance comparison between the proposed UniVAE and other VAEs on video reconstruction across WebVid-10M and Panda-70M validation set. We also present the video compression rate (VCR) and training parameters in Tab. 1. All the input videos are pre-processed to a length of 25 frames with resolution of $256 \times 256$ for evaluation. We select both 2D and 3D compressors as baselines for comparison. For 2D compressor, we choose VQ-GAN Esser et al. (2021), SD-VAE Rombach et al. (2022), and SVD-VAE Blattmann et al. (2023). For 3D compressor, TATS Ge et al. (2022), CV-VAE Zhao et al. (2024), OS-VAE Zheng et al. (2024), and OD-VAE Chen et al. (2024a) are utilized for comparison. As we can see, the proposed UniVAE achieves better superior reconstruction performance than all compared baselines among all three metrics across two validation datasets, while keep the same temporal ans spatial video compression ratio. We also present the qualitative comparison in Fig. 3.

We further evaluate different VAEs by connecting them with video diffusion models. As shown in Tab. 2, our UniVAE has better performance than CV-VAE and OD-VAE overall. Specifically, on the UCF101 dataset, Latte equipped with UniVAE achieves better FVD and KVD than that of the CV-VAE and that of the OD-VAE. Our UniVAE also brings better FVD on SkyTimelapse dataset, and has comparable KVD with OD-VAE. We also show the qualitative results in Fig. 4.

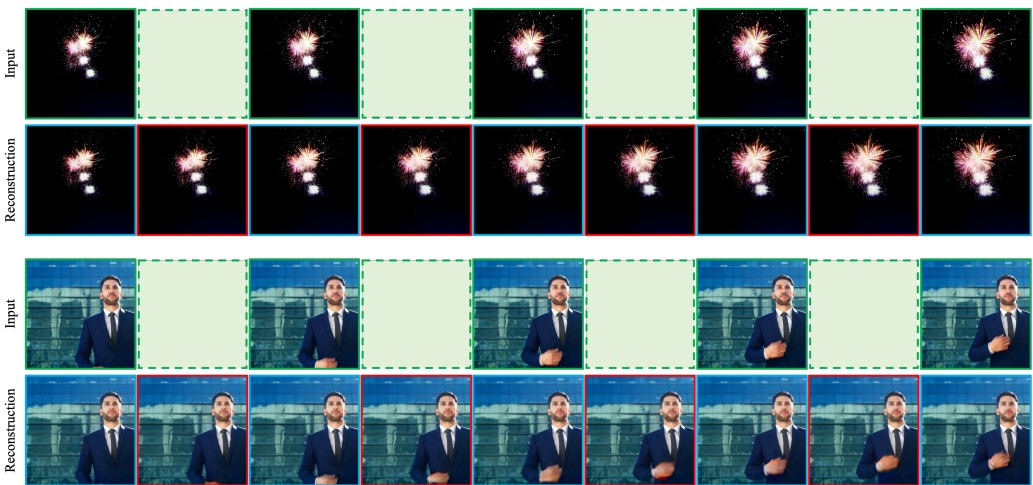

Figure 5: Reconstruction results of the refinement decoder $\mathcal{D}_2$.

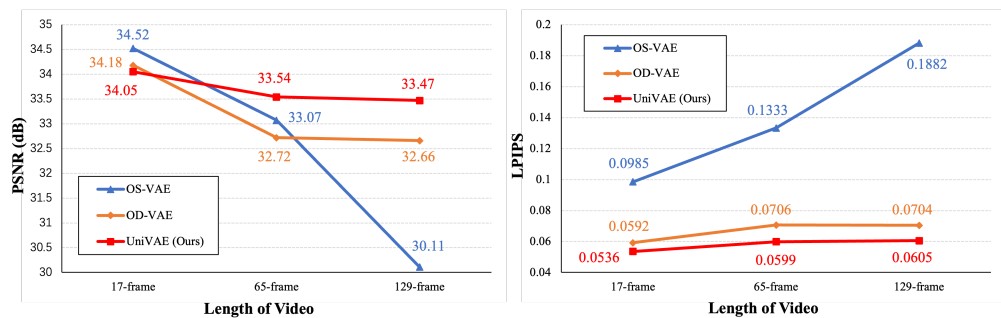

Figure 6: Performance (PSNR and LPIPS) comparison of different VAEs on video reconstruction across different frames.

### 4.3 EFFECTIVENESS OF THE REFINEMENT DECODER

In the proposed UniVAE, we introduce a refinement decoder $\mathcal{D}_2$ to interpolate frames with the assistance of decoder $\mathcal{D}_1$. To the best of our knowledge, the UniVAE is the first work to investigate the potential of VAE to interpolate reconstructed videos. It enables users to flexibly customize the generated final video output, *i.e.*, the user can choose between the ordinary videos reconstructed by $\mathcal{D}_1$ or the enhanced videos reconstructed by $\mathcal{D}_2$. As shown in Fig. 5, the images in green and blue boxes mean the input and reconstructed frames, respectively, while those in red box denote the generated intermediate frames. As we can see, the video sequences reconstructed by $\mathcal{D}_2$ exhibit greater smoothness compared with the input sequences. It is worth noting that except the proposed latent-guided refinement training strategy, we do not introduce any special module into $\mathcal{D}_2$. However, the generated additional frames still perform well. We attribute this to the multi-timescale latent representation extracted by $\mathcal{E}$, which greatly supports $\mathcal{D}_2$ to predict intermediate frames.

### 4.4 GENERALIZATION TO VIDEOS WITH DIFFERENT LENGTH

In this section, we discuss the generalization ability of VAEs to video sequences with different length. An ideal video VAE should be able to accurately reconstruct videos of any lengths. To evaluate the generalization of different video VAEs, we randomly select around 1,000 videos and clip them to different lengths: 17 frames for short videos, 65 frames for medium-length videos, and 129 frames for long videos. We use OS-VAE and OD-VAE as baselines for comparison since they have been used in popular LVDMs projects.

As shown in Fig. 6, all three VAEs achieve excellent performance in reconstructing relatively short videos (*i.e.*, 17-frame video reconstruction). However, with the length of input videos increasing, the performance of OS-VAE and OD-VAE drops significantly. For example, when the video re-

Table 3: Ablation results of UniVAE for video reconstruction on WebVid-10M validation set.

| Method | Settings of $\mathbf{F}$ | PSNR↑ | SSIM↑ | LPIPS↓ |
|---|---|---|---|---|
| Baseline | $\mathbf{F} = [3]$ | 31.16 | 0.8694 | 0.0586 |
| UniVAE-V | $\mathbf{F} = [3, 5]$ | 34.08 | **0.8785** | 0.0527 |
| UniVAE | $\mathbf{F} = [3, 5, 7, 9]$ | **34.13** | 0.8783 | **0.0525** |

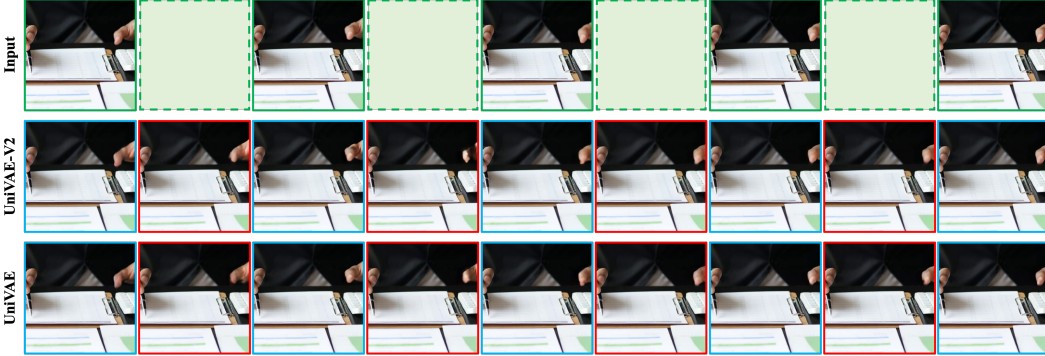

Figure 7: Qualitative comparison of the UniVAE and UniVAE-V2.

construction length increases from 17 frames to 129 frames, OS-VAE experiences a performance drop of 4.41dB/0.0897 in terms of PSNR/LPIPS. We analyze that this decline is due to its separate compression of temporal and spatial information. OD-VAE compresses videos across temporal and spatial dimensions simultaneously, which can capture spatial-temporal relationships better than OS-VAE. As we can see, OD-VAE has better generalization ability than OS-VAE, however, it still suffer obvious performance drop. In contrast, our UniVAE demonstrates superior generalization performance, with minimal decline as the length of input videos increases. We attribute it to our multi-scale temporal compression in $\mathcal{E}$, which encourages the UniVAE to capture the temporal patterns across various time-scale, and enables UniVAE to better adapt to videos with different length.

## 4.5 ABLATION STUDY

In this section, we perform comprehensive ablation studies to demonstrate the effectiveness of our designs in the proposed UniVAE. Specifically, we evaluate the benefits of multi-scale temporal downsampling module and the latent-guided refinement training strategy.

**Effectiveness of the Multi-Scale Temporal Downsampling Module.** To better capture the dynamic patterns across varying time scales in videos, we propose to utilize 3D convolution kernels with multiple kernel sizes to compress the video temporally. As described in Tab. 3, our UniVAE utilize 3D convolutions with four different kernel sizes (3, 5, 7, and 9) to perform temporal compression. We also design two variants for comparison. Among them, "UniVAE-V" uses only two different convolutions with kernel size 3 and 5. "Baseline" employs a single convolution with fixed size 3 for temporal compression, which is popular in current video VAEs Chen et al. (2024a). As we can see, UniVAE and UniVAE-V outperform "Baseline" by **2.97dB** and **2.92dB** in terms of PSNR, respectively, which suggests that using multi-scale convolutions is beneficial for temporal compression. On the other hand, we observe that UniVAE, with diverse set of convolutions, obtains slightly better performance than UniVAE-V. It indicates that increasing the diversity of convolutions in $\mathbf{F}$ can effectively capture temporal features in videos, thereby improving the reconstruction quality.

**Effectiveness of the Latent-Guided Refinement Training Strategy.** We propose a latent-guided refinement training strategy to train the standard decoder $\mathcal{D}_1$ and refinement decoder $\mathcal{D}_2$ in different stage. To verify the effectiveness of this, we propose a variant named UniVAE-V2, which directly trains $\mathcal{D}_2$ without the assistance of $\mathcal{D}_1$. As shown in Fig 7, we can see that our latent-guided refinement training strategy can better facilitate the additional frames generation in $\mathcal{D}_2$. The proposed latent-guided refinement training strategy not only ensures UniVAE standard video compression and reconstruction capability, (*i.e.*, $\mathcal{D}_1$), but also better encourages $\mathcal{D}_2$ to deliver smooth outputs.

## 5 CONCLUSIONS

In this study, we analyze the critical role of VAEs in current latent diffusion generative models, and present a unified video VAE for LVDMs, named `UniVAE`. We propose to utilize convolutions with varying kernel sizes in the encoder for temporal compression. Moreover, we explore the potential of VAE to generate additional frames to deliver smooth video content, and delicately design a latent-guided refinement training strategy for this purpose. Extensive experiments on video reconstruction and generation demonstrate the effectiveness of the `UniVAE`. We hope our work can motivate further research in VAE to advane the development of generative models.

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
