# OpenReview forum: "UniVAE: A Unified Frame-Enriched Video VAE for Latent Video Diffusion Models"
_ICLR.cc/2025/Conference — Submitted to ICLR 2025_

### Official Review · Reviewer_upgc · 2024-10-26

**Soundness:** 3
**Presentation:** 4
**Contribution:** 2
**Rating:** 3
**Confidence:** 5

**Summary:**

The paper introduces a unified video VAE for video diffusion models. It’s motivated by recent video diffusion models often relying on image-based VAEs, which only handle spatial compression and miss out on temporal compression. To solve this, the authors employ 3D convolutions to their video VAE, so it can compress both spatial and temporal information. They also include an interpolation decoder to improve smoothness in video output. The training process has two steps: first, they train the basic VAE encoder and decoder, then freeze them and train the interpolation decoder separately. They use Latte as the video diffusion model to evaluate UniVAE.

**Strengths:**

a. Clarity and Simplicity: The idea presented is straightforward, and the method section is generally easy to understand.
b. The experiments are comprehensive and convincing. The results demonstrate that the proposed method surpasses existing image-based VAE.
c. UniVAE includes an interpolation decoder that enables the generation of higher FPS videos.

**Weaknesses:**

1. Limited Novelty in Video VAE Design: Many recent video generation models, like CogVideoX, already use video VAEs that compress both spatial and temporal information. Hence the proposed UniVAE combining spatial and temporal compression is not novel to the field.
2. Interpolation Decoder Training: The interpolation decoder is trained on its own after freezing the basic VAE encoder and decoder, making it more like an independent interpolation model. It would be more meaningful if the basic VAE encoder and decoder and the interpolation decoder are trained together to improve the representation ability of latent space.
3. Interpolation Quality: The interpolation results shown in Fig.5 are less convincing, with obvious issues like loss of hand details. This further highlights the limitation of training the interpolation decoder separately.
4. Comparison with existing interpolation models: The paper lacks the comparison of UniVAE’s interpolation results with other interpolation methods. Adding this would give a clearer view of UniVAE’s effectiveness in generating higher frame rates and would address the concerns about it.

**Questions:**

1. The main limitation of this paper is the lack of novelty. The authors should explain how their VAE is different from current methods and what makes it stand out.
2. The authors need to show the real impact of the interpolation decoder and compare it with recent interpolation methods.

---

> ### Author Response · Authors · 2024-11-22
> **Response for Reviewer upgc**
>
> **Q1: Clarification on the novelty of the proposed UniVAE.**
>
> **A1:** This is an original effort to unify video reconsturction and frame interpolation in one VAE framework for latent video generative models, not on the spatial and temporal compression. There are two key innovations compared to existing video VAEs.
>
> (1) Multi-scale convolution kernels for temporal downsampling. We analyze the issues in existing video VAEs, and propose multi-scale temporal convolution kernels for temporal downsampling, which enhance the VAE's ability to capture dynamic patterns in videos, as described in Sec.3.2. It brings better video reconstruction and generation results, as shown in Tab.1 (main paper) and Tab.2 (main paper).
>
> (2) Frame interpolation within the UniVAE decoder. To the best of our knowledge, UniVAE is the first attempt to explore the potential of video VAEs to generate richer video content beyond reconstruction. We design the latent-guided refinement training strategy to enable UniVAE generate additional intermediate frames, which further allows latent video diffusion models to generate high fps videos with improved efficiency.
>
> **Q2: Interpolation Decoder Training: The interpolation decoder is trained on its own after freezing the basic VAE encoder and decoder, making it more like an independent interpolation model. It would be more meaningful if the basic VAE encoder and decoder and the interpolation decoder are trained together to improve the representation ability of latent space.**
>
> **A2:** (1) We do not agree the training of the refinement decoder is the same as training a separate interpolation model. Similar with Imagen [1] that utilizes super-resolution diffusion models to upsample the generated image, the refinement decoder is a component of the UniVAE. There are two key difference between the refinement decoder and frame interpolation method. (a) The frame interpolation methods typically can only generate the intermediate frame $x_{new}$ based on the given frames $[x_1, x_2]$. In contrast, our refinement decoder can directly reconstruct the video and the generated additional frame $[x_1, x_{new}, x_2]$ in a single pass. (b) The frame interpolation methods are typically independent and do not rely on video generation techniques. While, our refinement decoder generate video and additional frames based on the latent features extracted by the VAE encoder, which is integrated with our UniVAE.
>
> (2) If we train the UniVAE encoder $\mathcal{E}$, regular decoder $\mathcal{D}_1$, and refinement decoder $\mathcal{D}_2$ jointly, it may obscure the overall training objective of UniVAE. The optimization objective for video reconstruction in $\mathcal{D}_1$ and additional frame generation $\mathcal{D}_2$ may conflict, leading to unstable training.
>
> [1] Chitwan Saharia, William Chan, Saurabh Saxena, Lala Li, Jay Whang, Emily L Denton, Kamyar Ghasemipour, Raphael Gontijo Lopes, Burcu Karagol Ayan, Tim Salimans, et al. Photorealistic text-to-image diffusion models with deep language understanding. Advances in neural information processing systems, 35:36479–36494, 2022.

---

> > ### Author Response · Authors · 2024-11-22
> > **Response for Reviewer upgc**
> >
> > **Q3: Comparison with existing frame interpolation method.**
> >
> > **A3:** Here, we provide a analysis about the comparison between our UniVAE and existing frame interpolation method.
> >
> > (1) **Performance comparison:** We choose FILM [1] as baseline, and compare it with our UniVAE on frame interpolation. FILM is an independent frame interpolation method, which designs flow estimation modules to compute flows based on feature pyramids. For FILM, we first send the input video into the UniVAE and get the ordinary output from the regular decoder $\mathcal{D}_1$, and then use pre-trained FILM for frame interpolation. The input videos are pre-processed to a length of 65 frames with a resolution of $512 \times 512$. We show the qualitative results in Fig.9 (response letter). As we can see, our UniVAE achieve comparable interpolation results with the specifically designed interpolation method FILM.
> >
> > (2) **Time consumption:** We further provide the time consumption for the two pipelines, *i,e,*, UniVAE and "VAE + FILM". As described in (1), for each 65-frame video with a resolution of $512 \times 512$, we apply both interpolation methods to extend it to 129 frames, respectively. We show the time consumption of these two methods in Tab.1 below. Among them, "UniVAE" denotes we directly leverage the UniVAE to perform interpolation. The "VAE" and "FILM" mean UniVAE reconstruction and subsequent separate frame interpolation, respectively. As we can see, our UniVAE is more efficient than VAE + FILM, which further validate the potential of UniVAE in supporting latent video diffusion models.
> >
> > (3) **More discussion:** As the first attempt to utilize VAE for video interpolation, we focus on exploring the possibility and potential of video VAE for more frame generation. Compared to traditional independent interpolation methods, our UniVAE offers better coherence and integration. Moreover, the experiment results in (1) and (2) show that even without dedicated interpolation modules (such as flow estimation modules in FILM), our UniVAE still achieve comparable performance to exiting interpolation methods while reducing complexity of pipeline and improving efficiency. This is crucial for further advancing the latent video diffusion models.
> >
> > Table 1: Time consumption comparison between "UniVAE" and "VAE + FILM".
> > | Method | UniVAE | VAE (VAE + FILM) | FILM (VAE + FILM) | Total (VAE + FILM) |
> > | --- | --- | --- | --- | --- |
> > | Time Consumption | 22.83s | 6.72s | 464s | 470.72s |
> >
> > [1] Fitsum Reda, Janne Kontkanen, Eric Tabellion, Deqing Sun, Caroline Pantofaru, and Brian Curless. Film: Frame interpolation for large motion. In European Conference on Computer Vision, pp. 250–266. Springer, 2022.

---

> > > ### Comment · Reviewer_upgc · 2024-11-26
> > >
> > > Thanks for your response! After carefully reading the other reviewers' comments and the authors' replies, I still have some concerns about this method:
> > > 1. Adding an interpolation decoder directly is incremental and does not bring insight to this field. It doesn't provide meaningful improvements to the latent space. As shown in Fig. 5, the interpolation results lose details, making it less practical.
> > > 2. The comparison with FILM (a 2022 paper) isn’t very convincing since it's not the most recent interpolation method. This limits the relevance of the comparison.
> > > 3. The role of multi-scale convolution kernels VAE in video diffusion models seems unclear. The experiments were only conducted on Latte, which is a very weak baseline. To demonstrate the value of multi-scale convolution, the authors should test on stronger models like OpenSora or CogVideoX.

---

> > > > ### Author Response · Authors · 2024-11-26
> > > > **Response for Reviewer upgc**
> > > >
> > > > **Q1: More discussion about our interpolation decoder.**
> > > >
> > > > **A1:** To the best of our knowledge, UniVAE is the first attempt towards utilizing VAE for video interpolation. The UniVAE is proposed to unify video compression-reconstruction and interpolation to better support the latent video diffusion models. Compared to the traditional approach of separate reconstruction and interpolation, our UniVAE achieves task unification through model design, reducing reliance on independent modules for each task. This offers a new perspective for further advancing latent video diffusion models. On the other hand, the refinement decoder is not designed to improve the latent space. In contrast, it is designed to utilize the rich temporal information in the latent representations for additional intermediate frame generation. The multi-scale convolution kernels in encoder **E** is designed to improve the latent representation by capturing the dynamics across different time scales in videos.
> > > >
> > > > **Q2: More discussion about the comparison between UniVAE and existing frame interpolation methods.**
> > > >
> > > > **A2:** (1) As the original work to leverage VAE for generating additional frames, beyond existing video VAE, we focus on exploring the possibility and potential of using VAE decoder for frame interpolation. Thus, unlike current frame interpolation techniques, we do not design specific modules (like motion prediction module) in the refinement decoder, which may lead to detailed issues. However, as shown in Fig.5 (main paper), even without any specifically designed motion estimation modules, the intermediate frame generated by UniVAE can effectively capture the motion trajectory of the video, exhibiting strong global consistency and motion plausibility. This indicates that, despite some loss of detail, the generated frames successfully reflect the core motion characteristics of the video, which is a key objective in video interpolation tasks.
> > > >
> > > > (2) As we do not introduce any specific module in our UniVAE like existing frame interpolation techniques, it is not fair to compare our UniVAE with interpolation methods. On the other hand, our goal is not directly compete with specialized frame interpolation methods but to propose a novel framework that unifies video compression-reconstruction and interpolation within a single VAE, which can further advance the latent video diffusion models. This design is fundamentally unattainable with existing interpolation methods. Most current interpolation techniques focus solely on interpolation quality and often rely on explicit motion estimation. In contrast, our method eliminates the need for motion estimation module design, offering a completely new perspective. While the interpolation performance may fall short of SOTA interpolation methods, our work is the first to demonstrate the possibility of unifying video reconstruction and interpolation within in a single VAE, which provides a new direction for further advancing latent video diffusion models. Furthermore, the comparison with FILM in Table 8 (response letter) also demonstrate the potential efficiency of our design over interpolation methods.
> > > >
> > > > **Q3: Discussion about using Latte as the baseline.**
> > > >
> > > > **A3:** We aims to evaluate the performance of different VAEs to video generation tasks, rather than assessing the performance of diffusion models themselves. As this year's work, we consider Latte to be sufficiently stable and reliable to compare the performance of different VAEs.

---

> > > > > ### Comment · Reviewer_upgc · 2024-12-02
> > > > >
> > > > > Thanks for your response! I don't think the rebuttal addresses my concerns.
> > > > >
> > > > > 1. Directly training an interpolation decoder does not sufficiently demonstrate the novelty of the proposed method. Fundamentally, the interpolation decoder is essentially an upsample process, which can be easily implemented with training data. From a technical perspective, this offers limited insight or contribution to the field.
> > > > >
> > > > > 2. The authors argue that it is "unfair" to compare the interpolation decoder with recent interpolation methods, which is confusing. The proposed interpolation decoder makes no other contributions but is only used to implement interpolation. If the results exhibit significant detail loss and fail to reach the practical performance of recent interpolation methods, the contribution of this decoder becomes questionable. Without a clear advantage or practical relevance, it lacks meaningful value.
> > > > >
> > > > > 3. I think the authors should focus more on providing deeper insights to the field. For example, why does directly training an interpolation decoder lead to significant detail loss? Is it possible to address and repair this detail loss to achieve better performance? Simply positioning this as “the first attempt towards utilizing VAE for video interpolation” does not substantiate its value.
> > > > >
> > > > > 4. Evaluation on Latte is OK, but for a Unified Video VAE, relying on just one baseline is insufficient.  The authors should validate the proposed method on multiple models to better demonstrate its effectiveness and generalizability.

---

> > > > > > ### Author Response · Authors · 2024-12-03
> > > > > > **Response for Reviewer upgc**
> > > > > >
> > > > > > Thank you for your suggestions.
> > > > > >
> > > > > > We understand your concerns about the performance of our UniVAE for frame interpolation. In fact, at this stage, it is not realistic to directly use video VAE to achieve SOTA interpolation performance, and we also recognize this. However, we still want to explore the potential of video VAE for frame interpolation. If VAEs do have the potential for interpolation, then we may be able to establish a unified framework for video compression, reconstruction, and interpolation, which can further promote latent video diffusion models. Specific design like motion estimation can further be integrated into VAE decoder, which may achieve SOTA interpolation performance and improve efficiency as we show in Tab.8 (response letter).

---

### Official Review · Reviewer_kWyS · 2024-10-26

**Soundness:** 3
**Presentation:** 3
**Contribution:** 3
**Rating:** 6
**Confidence:** 4

**Summary:**

This paper explores how to design a better Video VAE for Video Latent Diffusion Models. Existing works either use image VAEs or video VAEs with simple single-scale convolution kernels, which fail to effectively model temporal consistency, thus requiring additional video frame interpolation models. This paper employs a multi-scale temporal convolution architecture across the channel dimension, helping the VAE Encoder capture dynamic patterns across different time scales without introducing additional computational cost. Furthermore, the authors propose a latent-guided refinement training scheme that integrates the frame interpolation function into the decoder to generate high frame rate videos, thereby producing videos with better temporal consistency, eliminate the need of extra video interpolation models. The effectiveness of the proposed method is demonstrated through experiments on video reconstruction and generation.

**Strengths:**

1. Using multi-scale temporal convolution to replace single-scale convolution is a reasonable approach. The multi-scale convolutions are designed across the channel dimension without introducing additional computational cost.
2. Introducing the frame interpolation function into the VAE decoder through latent-guided refinement training to improve temporal consistency in video generation and reconstruction is a very interesting idea.
3. The authors demonstrate the superiority of the proposed UniVAE on public datasets for both video reconstruction and generation.

**Weaknesses:**

1. The writing of the paper left me somewhat confused. In lines 79-80, what does "Our observations indicate..." refer to? How did the authors come to the subsequent conclusion that "If the encoder can effectively model temporal variations, the decoder could theoretically synthesize frames, leading to high fps videos without the for separate interpolation models"?
2. The paper emphasizes that both the Encoder and Decoder in UniVAE are causal. What specifically does "causal" mean in this context? What would be the results if either or both were non-causal?
3. Could the latent-guided refinement training scheme proposed in this paper also be used to enhance the performance of other pre-trained VAE?

**Questions:**

Please refer to the Weaknesses.

---

> ### Author Response · Authors · 2024-11-22
> **Response for Reviewer kWyS**
>
> **Q1: The writing of the paper left me somewhat confused. In lines 79-80, what does "Our observations indicate..." refer to? How did the authors come to the subsequent conclusion that "If the encoder can effectively model temporal variations, the decoder could theoretically synthesize frames, leading to high fps videos without the for separate interpolation models"?**
>
> **A1:** We apologize for the confusing expression in lines 79-80.
>
> (1) "Our observations" refer to the experiment results in Sec.4.2 and Sec.4.3. As we can see, our UniVAE not only fulfills the typical VAE role of video encoding and decoding in latent video diffusion models (Tab.1 (main paper) and Tab.2 (main paper)), but also extends to generate additional intermediate frames, as shown in Fig.5 (main paper). This is what we refer to in lines 79-80 as "VAEs can assume multiple roles in current video generation stream."
>
> (2) If the encoder of the VAE effectively captures the temporal dynamics of a video, the latent features $\mathbf{Z}$ will contain rich information about the sequence changes, such as object motion trajectories. This allows the decoder to leverage these dynamic patterns in $\mathbf{Z}$ to not only reconstruct the original video frames but also generate intermediate frames between them, leading to high fps videos without separate interpolation models. As we can see in Fig.5 (main paper), the refinement decoder can leverage the rich temporal information in $\mathbf{Z}$ to predict the motion trajectories and generate precise intermediate frames.
>
> **Q2: The paper emphasizes that both the Encoder and Decoder in UniVAE are causal. What specifically does "causal" mean in this context? What would be the results if either or both were non-causal?**
>
> **A2:** The term ``causal" indicates that the VAE processes the relationships between frames in a causal manner when encoding and decoding video. Specifically, when handling frame $f_i$, it only considers the current frame $f_i$ and the preceding frames $[f_1, f_2, ..., f_{i-1}]$, without referencing subsequent frames, which means the output for each frame only depends on the previous frames. This design enables VAE to encode both video and image [1]. The causal setup is widely applied in decoder-only architectures of large language models (LLMs) and video VAEs [2][3][4].
>
> [1] Lijun Yu, Jos´e Lezama, Nitesh B Gundavarapu, Luca Versari, Kihyuk Sohn, David Minnen, Yong Cheng, Agrim Gupta, Xiuye Gu, Alexander G Hauptmann, et al. Language model beats diffusion–tokenizer is key to visual generation. arXiv preprint arXiv:2310.05737, 2023.
>
> [2] Zangwei Zheng, Xiangyu Peng, Tianji Yang, Chenhui Shen, Shenggui Li, Hongxin Liu, Yukun Zhou, Tianyi Li, and Yang You. Open-sora: Democratizing efficient video production for all, March 2024. URL https://github.com/hpcaitech/Open-Sora.
>
> [3] Sijie Zhao, Yong Zhang, Xiaodong Cun, Shaoshu Yang, Muyao Niu, Xiaoyu Li, Wenbo Hu, and Ying Shan. Cv-vae: A compatible video vae for latent generative video models. arXiv preprint arXiv:2405.20279, 2024.
>
> [4] Liuhan Chen, Zongjian Li, Bin Lin, Bin Zhu, Qian Wang, Shenghai Yuan, Xing Zhou, Xinghua Cheng, and Li Yuan. Od-vae: An omni-dimensional video compressor for improving latent video diffusion model. arXiv preprint arXiv:2409.01199, 2024.
>
> **Q3: Could the latent-guided refinement training scheme proposed in this paper also be used to enhance the performance of other pre-trained VAE?**
>
> **A3:** Yes. The latent-guided refinement training scheme can be utilized in other pre-trained video VAE. Here, we retrain OD-VAE with our proposed latent-guided refinement training scheme, and show the qualitative comparison in Fig.8 (response letter). As we an see, our latent-guided can also be utilized to enhance the output of OD-VAE. While, our UniVAE still obtains better results.

---

> > ### Comment · Reviewer_kWyS · 2024-11-25
> >
> > Thank you for your response. My concerns have been addressed by the authors in their rebuttal, so I will maintain the current score.

---

> > > ### Author Response · Authors · 2024-12-03
> > > **Response for Reviewer kWyS**
> > >
> > > Thank you for your recognition of our work.

---

### Official Review · Reviewer_nohp · 2024-11-03

**Soundness:** 3
**Presentation:** 3
**Contribution:** 3
**Rating:** 6
**Confidence:** 4

**Summary:**

The paper introduces two key improvements to the VAE model architecture: the Multi-Scale Spatial-Temporal Downsampling method, which captures information across different temporal scales to address the existing flickering issue in VAEs, and the Latent-Guided Refinement Training approach, which explores the potential of interpolating within the VAE's latent space. The authors provide both quantitative and qualitative analyses of the proposed VAE's performance, demonstrating commendable results.

**Strengths:**

1. The usage of Multi-Scale Spatial-Temporal Downsampling is good, as it effectively captures content across different temporal scales. The accompanying ablation experiments convincingly demonstrate that this downsampling approach yields substantial improvements in the performance metrics.
2. This paper explores the potential of latent space for frame interpolation, providing a valuable perspective and reference for future research.
3. The study also investigates the degradation of current VAEs when video duration is extended, offering a noteworthy analysis angle.

**Weaknesses:**

The effectiveness of the refinement decoder is not sufficiently demonstrated, as it relies solely on the qualitative analysis presented in Figure 5. However, I have observed a color bleeding phenomenon in the interpolated images. While this is a good idea, I would appreciate more discussion on this aspect.

**Questions:**

1. Given that most current VAEs utilize tiling to reduce inference memory, I wonder if the Multi-Scale Spatial-Temporal downsampling method imposes any limitations within tiling. Can it still ensure satisfactory performance in such scenarios?
2. Have the authors attempted to use latent videos generated by diffusion models (e.g., Latte) as input to the refinement decoder? This approach may pose greater challenges compared to directly encoding videos and performing interpolation, but it could also provide stronger evidence for the significance of the proposed method.

---

> ### Author Response · Authors · 2024-11-22
> **Response for Reviewer nohp**
>
> **Q1: The effectiveness of the refinement decoder is not sufficiently demonstrated, as it relies solely on the qualitative analysis presented in Figure 5. However, I have observed a color bleeding phenomenon in the interpolated images. While this is a good idea, I would appreciate more discussion on this aspect.**
>
> **A1:** (1) As the original work to leverage VAE for generating additional frames, beyond existing video VAE, we focus on exploring the possibility and potential of using VAE decoder for frame interpolation. Thus, unlike current frame interpolation techniques, we do not design specific modules (like motion prediction module) in the refinement decoder, which may lead to detailed issues such as color bleeding in the generated intermediate frames. However, as shown in Fig.5 (main paper), even without any specially designed motion estimation modules, our UniVAE can still accurately predict the motion trajectories in videos, and generate precise intermediate frames.
>
> (2) To further prove the effectiveness of our UniVAE for frame interpolation, we compare our UniVAE with existing separate interpolation method FILM [1]. FILM is an independent frame interpolation method, which designs flow estimation modules to compute flows based on feature pyramids. For FILM, we first send the input video into the UniVAE and get the ordinary output from the regular decoder $\mathcal{D}_1$, and then use pre-trained FILM for frame interpolation. We denote it as ``VAE + FILM''.
>
> (a) **Performance comparison:** The input videos are pre-processed to a length of 65 frames with a resolution of $512 \times 512$. We show the qualitative results in Fig.5 (response letter). As we can see, our UniVAE achieve comparable interpolation results with the specifically designed interpolation method FILM.
>
> (b) **Time consumption:** We further provide the time consumption for the two pipelines, *i,e,*, UniVAE and VAE + FILM. As described in (1), for each 65-frame video with a resolution of $512 \times 512$, we apply the both interpolation methods to extend it to 129 frames, respectively. We show the time consumption of these two methods in Tab.1 below. As we can see, video reconstruction is faster than video interpolation in UniVAE. However, the subsequent separate interpolation will cost more time than UniVAE. As a result, our UniVAE is more efficient than "VAE + FILM", which further prove the potential of UniVAE in supporting latent video diffusion models.
>
> The experiment results in Fig.5 (response letter) and Tab.1 below show that even without dedicated interpolation modules (such as flow estimation modules in FILM), our UniVAE still achieve comparable performance to exiting interpolation methods while reducing complexity of pipeline and improving efficiency. This is crucial for further advancing the latent video diffusion models.
>
> Table 1: Time consumption comparison between UniVAE and "VAE + FILM".
>
> | Method | UniVAE | VAE (VAE + FILM) | FILM (VAE + FILM) | Total (VAE + FILM) |
> | --- | --- | --- | --- | --- |
> | Time Consumption | 22.83s | 6.72s | 464s | 470.72s |
>
> [1] Fitsum Reda, Janne Kontkanen, Eric Tabellion, Deqing Sun, Caroline Pantofaru, and Brian Curless. Film: Frame interpolation for large motion. In European Conference on Computer Vision, pp. 250–266. Springer, 2022.
>
>
> **Q2: Given that most current VAEs utilize tiling to reduce inference memory, I wonder if the Multi-Scale Spatial-Temporal downsampling method imposes any limitations within tiling. Can it still ensure satisfactory performance in such scenarios?**
>
> **A2:** Our spatial-temporal downsampling imposes no limitations within tiling. In fact, following the previous methods [2], we also employ tiling technique when evaluating the performance of our UniVAE, and show the results in Tab.1 (main paper), Tab.2 (main paper), and Fig.6 (main paper).
>
> [2] Liuhan Chen, Zongjian Li, Bin Lin, Bin Zhu, Qian Wang, Shenghai Yuan, Xing Zhou, Xinghua Cheng, and Li Yuan. Od-vae: An omni-dimensional video compressor for improving latent video diffusion model. arXiv preprint arXiv:2409.01199, 2024.
>
> **Q3: Have the authors attempted to use latent videos generated by diffusion models (e.g., Latte) as input to the refinement decoder? This approach may pose greater challenges compared to directly encoding videos and performing interpolation, but it could also provide stronger evidence for the significance of the proposed method.**
>
> **A3:** Thanks for your good suggestion. Since the regular decoder $\mathcal{D}_1$ and the refinement decoder $\mathcal{D}_2$ share the latent representation $\mathbf{Z}$, the Latte equipped with our UniVAE can directly leverage the refinement decoder to output videos with richer content. We show some qualitative results in Fig.6 (response letter).

---

### Official Review · Reviewer_Dx2p · 2024-11-04

**Soundness:** 2
**Presentation:** 3
**Contribution:** 2
**Rating:** 3
**Confidence:** 5

**Summary:**

This paper presents a video VAE that compresses input videos both spatially and temporally. The key contributions are twofold. First, unlike most previous approaches that use a fixed temporal kernel size for downsampling, the authors propose a multi-kernel temporal convolution scheme to better capture the video’s temporal dynamics. Second, they incorporate an interpolation decoder alongside a reconstruction decoder, conditioned on both the sampled latent and the reconstructed outputs, enabling flexible video generation. Experimental results are provided for both video reconstruction and generation tasks.

**Strengths:**

+ The multi-kernel convolution approach in the encoder network is intriguing and should be further explored
+ Integrating an interpolation module within a VAE model is novel and could provide a more efficient approach for unified video compression and generation.

+ The quantitative results are strong.
+ The paper is generally well-written.

**Weaknesses:**

- *The various ideas introduced in the paper lack sufficient motivation, and the technical contribution is limited.*

    - The introduction does not convincingly explain why we need the proposed multi-kernel convolution approach.

   - Why does temporal pooling operation limit the ability of video VAEs (Line 51)? Why would applying different kernels across channel partitions necessarily facilitate better temporal modeling? How did the authors decide to apply the kernels for each channel partition? Which partition uses the smaller temporal kernels and the larger ones?

    - Why is the multi-kernel convolution applied only during downsampling? Did the authors employ the same scheme in the upsampling stage? As the spatiotemporal dimension has to match across the partitions, a larger temporal kernel also implies using larger temporal padding and stride which could be counterintuitive to the basic benefit the authors are arguing.

    - If the interpolation decoder is trained independently, isn't it the same as training a separate interpolation model (given that the encoder and the reconstruction decoder are frozen)? How important is it to use the sampled latent as an input to the interpolation decoder? What is the benefit of adding the reconstructed frames in the interpolation decoder? How does the overall approach compare with just using a pre-trained, state-of-the-art interpolation model on top of the reconstructed frames? There should be a more thorough analysis in this regard.

    - The claim that this is the first paper to unify spatial and temporal modeling in the encoder is too strong and not correct (Line 111). Many previous works such as OS-VAE and OD-VAE do both spatial and temporal compression.

    - The authors claim their work is specifically for video data (Line 106). Then, what is the motivation for using causal 3D convolutional networks which are commonly used for joint image and video encoding/decoding? Why formulate the problem using N+1 frames?


- *Several important details are missing, making it challenging to assess whether a fair experimental comparison was conducted.*

    - What is the dataset used for training the proposed VAE? Why use 10^6 steps for the second stage? Isn’t it too much? What are the coefficients used for the different loss functions? Did the authors train their model from scratch or initialize it with pre-trained weights from image VAEs? Did the GAN training start at the same time as the reconstruction training in stage 1? Do both stage 1 and stage 2 use the same set of loss functions?

    - The resolution of the videos used for evaluation in Table 1 is quite small (x256). Why? Why not try the inference at a higher resolution if the design is GPU-efficient?

    - For the competing baselines, did the authors use pre-trained checkpoints or train from scratch? I understand it is not an easy task to train the other models from scratch and I don’t expect that. However, given the significant leap in reconstruction performance (+3dB), I want to hear more about the efforts the authors put on to ensure the comparisons are as fair as they can be.

    - Why is the FVD score so high in Table 2? What is the length of the video clips used for evaluation? How many clips are generated? The numbers indicate poor generation quality (i.e. very high FVD). Isn’t it counterintuitive to the reconstruction performance of the decoder? What are the settings used in generation training and evaluation?


- *The experimental results and ablations are not thorough enough to fully demonstrate the proposed ideas' merits.*

    - The authors fail to convincingly argue why their proposed approach gives a very strong performance compared to previous methods in Section 4.2. What makes the proposed approach very unique to achieve such a performance? Does using multiple kernels give such a performance boost?

    - The qualitative analysis in Fig. 4 is not very informative. An example depicting the differences between the different methods would have been better.

    - The claim in Lines 419-420 does not hold unless a comparison with a pre-trained interpolation method is shown

    - Which video dataset is used for the experiment in Section 4.4? The performance gap between OD-VAE and UniVAE in Fig. 6 does not seem to be large for small (17) or medium (65) length frames. This appears to be contradictory to the results reported in Table 1 which are evaluated at 25 frames. Why is that?

    - The ablation results in Fig 7 are not convincing. It can be argued that the results of UniVAE-v2 seem to be better (sharper details)  than UniVAE

- *The qualitative analysis is limited.*

    - There are no video results for either reconstruction or generation tasks. This is critical to visualize the benefit of the proposed design choices beyond quantitative numbers.

    - The overall qualitative analysis in the paper is also quite limited

**Questions:**

Please refer to the questions (or issues) mentioned in the Weaknesses section.

---

> ### Author Response · Authors · 2024-11-22
> **Response for Reviewer Dx2p**
>
> **1. Clarification on our motivation and technical contribution.**
>
> We sincerely appreciate your insightful comments and will revise our wording and discussion accordingly. In our humble opinion, joint spatiotemporal modeling using multi-scale convolution in video VAE is a reasonable and intriguing approach for both diffusion and interpolation, which may be beneficial to share with the field timely.
>
> **Q1: The introduction does not convincingly explain why we need the proposed multi-kernel convolution approach.**
>
> **A1:** Thanks. The challenges of handling video data largely come from their multi-resolution nature in both spatial and temporal domain, which has been the research theme for multidecade. The multi-scale convolution with a large receptive field in VAE offers one way to enhance compression and reconstruction quality of *long-duration* and *high frame rate* videos, especifically using joint spatiotemporal modeling at the feature level in one framework for both diffusion and interpolation tasks. We would not claim multi-scale convolution is the only way, yet we claim it deserves the research efforts for its potential in VAE and video regeneration.
>
> From the perspective of experiments results:
>
> (1) Single-scale convolution kernel results in video reconstruction performance degradation. As illustrated in Fig.6 (main paper), both OS-VAE and OD-VAE which use single-scale convolution kernel exhibit performance degradation when reconstructing long videos, due to the limited ability of single-scale convolution kernel to capture changes occurring over varying time scales in videos.
>
> (2) Single-scale convolution cannot support our refinement decoder design. Compared with single-scale convolution, multi-scale convolution kernel can better extract temporal patterns at various time scales, which can provide the refinement decoder more informative features for additional intermediate frame generation. Our multi-scale convolution kernels are indispensable for the subsequent refinement decoder. To demonstrate the advantage of multi-scale convolution kernels for additional frame generation, we train a new refinement decoder based on pre-trained OD-VAE that only uses single-scale convolution kernel when temporal downsampling. The comparison in Fig.2 (response letter) shows UniVAE with multi-scale convolution kernels can generate better intermediate frames compared with OD-VAE with single-scale convolution.
>
> From the perspective motivation:
>
> We aims to improve VAE performance by enhancing the encoder's ability to capture temporal clues for spatio-temporal modeling. Our multi-scale temporal convolution kernels endows the UniVAE ability to perceive and capture the dynamics patterns across different time scales in videos, which can result in better video reconstruction and generation results compared with existing video VAE, as shown in Tab.1 (main paper) and Tab.2 (main paper).
>
> **Q2: Why does temporal pooling operation limit the ability of video VAEs (Line 51)?**
>
> **A2:** The temporal pooling operation is a rudimentary approach to temporal dimensionality reduction, as evidenced by various improved methods [1] developed for different tasks [2][3] over time. Fixed pooling operations may miss visual or motion details and lack of learnability. Instead, 3D convolutions generally are robust to capture more appearance details and varying motion patterns [4]. To further support this argument, we replace the temporal downsampling operator in our UniVAE with average pooling, and shows the comparison results in Tab.1 below. As we can see, our UniVAE achieves better performance than VAE with temporal pooling operator.
>
> Table 1. Performance comparison between different VAEs on video reconstruction.
> | Method | PSNR (17-frame) | LPIPS (17-frame) | PSNR (65-frame) | LPIPS (65-frame) | PSNR (129-frame) | LPIPS (129-frame) |
> |---|---|---|---|---|---|---|
> | Pooling-VAE | 33.91 | 0.0574 | 33.48 | 0.0621 | 33.41 | 0.0623 |
> | UniVAE | 34.05 | 0.0536 | 33.54 | 0.0599 | 33.47 | 0.0605 |
>
> [1] Avraham Ruderman, Neil C Rabinowitz, Ari S Morcos, and Daniel Zoran. Pooling is neither necessary nor sufficient for appropriate deformation stability in cnns. arXiv preprint arXiv:1804.04438, 2018.
>
> [2] Eric Kauderer-Abrams. Quantifying translation-invariance in convolutional neural networks. arXiv preprint arXiv:1801.01450, 2017.
>
> [3] Jonathan Long, Evan Shelhamer, and Trevor Darrell. Fully convolutional networks for semantic segmentation. In Proceedings of the IEEE conference on computer vision and pattern recognition, pp. 3431–3440, 2015.
>
> [4] Lijun Yu, Jos´e Lezama, Nitesh B Gundavarapu, Luca Versari, Kihyuk Sohn, David Minnen, Yong Cheng, Agrim Gupta, Xiuye Gu, Alexander G Hauptmann, et al. Language model beats diffusion–tokenizer is key to visual generation. arXiv preprint arXiv:2310.05737, 2023.

---

> > ### Author Response · Authors · 2024-11-22
> > **Response for Reviewer Dx2p**
> >
> > **Q3: Why would applying different kernels across channel partitions necessarily facilitate better temporal modeling? How did the authors decide to apply the kernels for each channel partition? Which partition uses the smaller temporal kernels and the larger ones?**
> >
> > **A3:** (1) The multi-scale convolution kernels provide varying receptive fields, enabling the model to capture temporal modeling across different scales. This statement is supported by evidence from [1]. Moreover, our ablation study in Tab. 3 (main paper) also supports this point.
> >
> > (2) The application of channel partitions is to reduce the computation cost. In fact, we can re-design several new convolution kernels $F = [f^{new}_1, f^{new}_2, ..., f^{new}_p]$, and directly apply them to the feature $\mathbf{x} \in \mathbb{R}^{N \times H \times W \times C}$ without channel partition, then fuse them together and get the final result $\mathbf{y} = \textit{fusion}(f^{new}_1 \otimes \mathbf{x}, f^{new}_2 \otimes \mathbf{x}, ..., f^{new}_p \otimes \mathbf{x})$. However, compared with single-scale convolution, this design introduces $(p - 1)$ additional convolution kernels, increasing the computational load. To make a better trade-off between effectiveness and efficiency, we draw inspiration from [2] and choose to apply different convolution kernels across channel partitions as described in Sec.3.2. The multi-scale convolution kernels can encourage VAE to capture the dynamic patterns across different time scales, while the application of channel partition reduces the parameters of each convolution kernel, to lower the overall computational cost of VAE.
> >
> > [1] Xiaolong Wang, Ross Girshick, Abhinav Gupta, and Kaiming He. Non-local neural networks. In Proceedings of the IEEE conference on computer vision and pattern recognition, pp. 7794–7803, 2018.
> >
> > [2] Xiaohan Ding, Yiyuan Zhang, Yixiao Ge, Sijie Zhao, Lin Song, Xiangyu Yue, and Ying Shan. Unireplknet: A universal perception large-kernel convnet for audio video point cloud time-series and image recognition. In Proceedings of the IEEE/CVF Conference on Computer Vision and Pattern Recognition, pp. 5513–5524, 2024.
> >
> > **Q4: Why is the multi-kernel convolution applied only during downsampling? Did the authors employ the same scheme in the upsampling stage? As the spatiotemporal dimension has to match across the partitions, a larger temporal kernel also implies using larger temporal padding and stride which could be counterintuitive to the basic benefit the authors are arguing.**
> >
> > **A4:** We do not employ the multi-kernel convolution in the upsampling since the enhanced encoded latent representations are sufficient to reconstruct a video with simple upsampling. Below, we provide a detailed explanation along with supplementary experimental evidence.
> >
> > In VAE, the role of encoder $\mathcal{E}$ is to map input videos to the latent representation $\mathbf{Z}$. When performing temporal compression, the downsampling operation inevitably leads to information loss. To address this, we introduce multi-scale kernels for temporal downsampling. This allows $\mathcal{E}$ to capture temporal dynamics at multiple time scales, reducing information loss caused by single-scale downsampling and improving the quality of the latent representation. Compared to the decoder, the encoder plays a more critical role, as it compresses the video into the latent representation that will be utilized to train the latent video diffusion models. If too much information is lost during downsampling, the decoder will struggle to reconstruct high-quality video.
> >
> > On the other hand, decoder is responsible for reconstructing input videos from the latent representation $\mathbf{Z}$ produced by $\mathcal{E}$. Although using multi-scale kernels in the decoder may enhance the reconstruction quality, our primary focus is on the encoder, as the quality of the encoded latent representation directly impacts the subsequent diffusion models. Therefore, we prioritize applying multi-scale kernels in encoder.
> >
> > We further design a variant that adopt multi-scale convolution kernels on both downsampling and upsampling modules, which is denoted as ``UniVAE-V3''. In Tab.2 below, the results show that given the improved latent space obtained by multi-scale convolutions, simple upsampling can do a fairly good job to reconstruct the video in terms of PSNR and LPIPS.
> >
> > Table 2: Performance comparison between different VAEs on video reconstruction.
> > | Method | PSNR (17-frame) | LPIPS (17-frame) | PSNR (65-frame) | LPIPS (65-frame) | PSNR (129-frame) | LPIPS (129-frame) |
> > | --- | --- | --- | --- | --- | --- | --- |
> > | UniVAE-V3 | 33.68 | 0.0614 | 32.98 | 0.0731 | 32.88 | 0.0753 |
> > | UniVAE | 34.05 | 0.0536 | 33.54 | 0.0599 | 33.47 | 0.0605 |

---

> > > ### Author Response · Authors · 2024-11-22
> > > **Response for Reviewer Dx2p**
> > >
> > > **Q5: If the interpolation decoder is trained independently, isn't it the same as training a separate interpolation model (given that the encoder and the reconstruction decoder are frozen)?**
> > >
> > > **A5:** We do not agree the training of the refinement decoder is the same as training a separate interpolation model. Similar with Imagen [1] that utilizes super-resolution diffusion models to upsample the generated image, the refinement decoder is a component of the UniVAE. There are two key difference between the refinement decoder and a standalone frame interpolation method. (a) The frame interpolation methods typically can only generate the intermediate frame $x_{new}$ based on the given frames $[x_1, x_2]$. In contrast, our refinement decoder can directly reconstruct the video and the generated additional frame $[x_1, x_{new}, x_2]$ in a single pass. (b) The frame interpolation methods are typically independent and do not rely on video generation techniques. In contrast, our refinement decoder generate video and additional frames based on the latent features extracted by the VAE encoder, which is integrated with our UniVAE.
> > >
> > > [1] Chitwan Saharia, William Chan, Saurabh Saxena, Lala Li, Jay Whang, Emily L Denton, Kamyar Ghasemipour, Raphael Gontijo Lopes, Burcu Karagol Ayan, Tim Salimans, et al. Photorealistic text-to-image diffusion models with deep language understanding. Advances in neural information processing systems, 35:36479–36494, 2022.
> > >
> > > **Q6: What is the benefit of adding the reconstructed frames in the interpolation decoder?**
> > >
> > > **A6:** As described in Sec.3.3, adding the reconstructed frames ($\mathcal{D}_1$) in the interpolation decoder ($\mathcal{D}_2$), reducing $\mathcal{D}_2$'s burden of reconstructing existing frames and allowing $\mathcal{D}_2$'s to focus more on generating additional frames (line 264-267). Also, we demonstrate the effectiveness of this design in Fig.7 in ablation study. As we can see, our latent-guided refinement training strategy can better preserve hand details (specific in Fig.4 (reponse letter)).
> > >
> > > **Q7: How does the overall approach compare with just using a pre-trained, state-of-the-art interpolation model on top of the reconstructed frames? There should be a more thorough analysis in this regard.**
> > >
> > > **A7:** Here, we provide a thorough analysis with existing frame interpolation method.
> > >
> > > (1) **Performance Comparison:** We choose FILM [2] as baseline, and compare it with our UniVAE on frame interpolation. FILM is an independent frame interpolation method, which designs flow estimation modules to compute flows based on feature pyramids. For FILM, we first send the input video into the UniVAE and get the ordinary output from the regular decoder $\mathcal{D}_1$, and then use pre-trained FILM for frame interpolation. The input videos are pre-processed to a length of 65 frames with a resolution of $512 \times 512$. We show the qualitative results in Fig.3 (response letter). As we can see, our UniVAE achieve comparable interpolation results with the specifically designed interpolation method FILM.
> > >
> > > (2) **Time Consumption:** We further provide the time consumption for the two pipelines, *i.e.*, UniVAE and VAE + FILM. As described in (1), for each 65-frame video with a resolution of $512 \times 512$, we apply both interpolation methods to extend it to 129 frames, respectively. We show the time consumption of these two methods in Tab.3 below. Among them, UniVAE denotes we directly leverage the UniVAE to perform interpolation. VAE and FILM mean UniVAE reconstruction and subsequent separate frame interpolation, respectively. As we can see, our UniVAE is more efficient than VAE + FILM, which further prove the potential of UniVAE in supporting latent video diffusion models.
> > >
> > > Table 3: Time consumption comparison between UniVAE and VAE + FILM.
> > >
> > > | Method | UniVAE | VAE (VAE + FILM) | FILM (VAE + FILM) | Total (VAE + FILM) |
> > > | --- | --- | --- | --- | --- |
> > > | Time Consumption | 22.83s | 6.72s | 464s | 470.72s |
> > >
> > > (3) **More discussion:** As the first attempt to utilize VAE for video interpolation, we focus on exploring the possibility and potential of video VAE for more frame generation. Compared to traditional independent interpolation methods, our UniVAE offers better coherence and integration. Moreover, the experiment results in (1) and (2) show that even without dedicated interpolation modules (such as flow estimation modules in FILM), our UniVAE still achieve comparable performance to exiting interpolation methods while reducing complexity of pipeline and improving efficiency. This is useful for further advancing the latent video diffusion models.
> > >
> > > [2] Fitsum Reda, Janne Kontkanen, Eric Tabellion, Deqing Sun, Caroline Pantofaru, and Brian Curless. Film: Frame interpolation for large motion. In European Conference on Computer Vision, pp. 250–266. Springer, 2022.

---

> > > > ### Author Response · Authors · 2024-11-22
> > > > **Response for Reviewer Dx2p**
> > > >
> > > > **Q8: (1) The claim that this is the first paper to unify spatial and temporal modeling in the encoder is too strong and not correct (Line 111). Many previous works such as OS-VAE and OD-VAE do both spatial and temporal compression. (2) The authors claim their work is specifically for video data (Line 106). Then, what is the motivation for using causal 3D convolutional networks which are commonly used for joint image and video encoding/decoding? Why formulate the problem using N+1 frames?**
> > > >
> > > > **A8:** We apologize for our unclear and confusing wording. Here, we clarify them.
> > > >
> > > > **(1) The claim in Line 111:** Spatiotemporal compression does not equate to unify spatial and temporal modeling, as there are various methods to achieve spatiotemporal compression, such as tandem VAE (*e.g*., OS-VAE), OD-VAE uses 3D convolution for spatial and temporal modeling. We will revise it and update the sentence to ``multi-scale spatiotemporal modeling in the encoder''.
> > > >
> > > > **(2) The claim in Line 106:** Causal 3D Convolutional Networks are widely utilized in video VAEs, which allows them to encode/decode image and video jointly. However, in this paper, we focus on their capability in videos, aligning with the motivations of some video VAE works like [1][2][3]. Additionally, the *``N+1 frames''* is a common practice in VAEs, particularly in video diffusion models, and we have adopted it in our work. We will revise the expression to make it clearer.
> > > >
> > > > [1] Zhuoyi Yang, Jiayan Teng, Wendi Zheng, Ming Ding, Shiyu Huang, Jiazheng Xu, Yuanming Yang, Wenyi Hong, Xiaohan Zhang, Guanyu Feng, et al. Cogvideox: Text-to-video diffusion models with an expert transformer. arXiv preprint arXiv:2408.06072, 2024.
> > > >
> > > > [2] Zangwei Zheng, Xiangyu Peng, Tianji Yang, Chenhui Shen, Shenggui Li, Hongxin Liu, Yukun Zhou, Tianyi Li, and Yang You. Open-sora: Democratizing efficient video production for all, March 2024. URL https://github.com/hpcaitech/Open-Sora.
> > > >
> > > > [3] Liuhan Chen, Zongjian Li, Bin Lin, Bin Zhu, Qian Wang, Shenghai Yuan, Xing Zhou, Xinghua Cheng, and Li Yuan. Od-vae: An omni-dimensional video compressor for improving latent video diffusion model. arXiv preprint arXiv:2409.01199, 2024.
> > > >
> > > >
> > > >
> > > > **2. Clarification on experimental details.**
> > > >
> > > > **More training steps in stage 2.** In stage 2, we aim to train the refinement decoder to estimate the additional intermediate frames based on the latent feature Z, which is more difficult than the reconstruction in stage 1. So we train the refinement decoder for more steps in stage 2.
> > > >
> > > > **The coefficients used for different loss functions.** Following the common setting in previous video VAE training [1], we set the cofficients of reconstruction loss, adversarial loss, and KL regularization as 1, 5000, and 1e-06, respectively.
> > > >
> > > > **GAN training.** The GAN training starts after 2000 steps in stage 1.
> > > >
> > > > **The loss functions in stage 1 and stage 2.** Both stage 1 and stage 2 use the same set of loss functions in Eq.1. While, the encoder $\mathcal{E}$ and the regular decoder $\mathcal{D}_1$ are frozen in this process.
> > > >
> > > > **The $256 \times 256$ resolution video for evaluation.** For fair comparison, we follow the common settings in previous works [1][2], which typically transform videos to clips of 25-frame length and $256 \times 256$ resolution for evaluation.
> > > >
> > > > **Training dataset and Model Initialization.** (1) Training dataset: We collect 1.5M videos from Internet, and use them to train our UniVAE. Since most of the data for training VAEs is private and unavailable [1][2], it is difficult to set them uniformly. (2) Model Initialization: For training efficiency, we initialize part of modules in our UniVAE with parameters from OD-VAE to provide UniVAE with an initial video encoding and decoding capability, while the other part of modules are initialized randomly.
> > > >
> > > > **Clarification on video generation performance and details.** The high FVD values in Tab.2 are due to the insufficient training of the latent diffusion model (Latte), which is not counterintuitive to the reconstruction performance of VAEs. When training the latent diffusion model, we train Latte equipped with different VAEs on UCF101 and SkyTimelapse for 100K steps, due to the time and computational constraints. Each training sample is pre-processed to a length of 24 frames with a resolution of $256 \times 256$. During evaluation, we generate 100 samples with 24-frame length and $256 \times 256$ resolution to calculate the FVD and KVD metrics.
> > > >
> > > > [1] Liuhan Chen, Zongjian Li, Bin Lin, Bin Zhu, Qian Wang, Shenghai Yuan, Xing Zhou, Xinghua Cheng, and Li Yuan. Od-vae: An omni-dimensional video compressor for improving latent video diffusion model. arXiv preprint arXiv:2409.01199, 2024.
> > > >
> > > > [2] Zangwei Zheng, Xiangyu Peng, Tianji Yang, Chenhui Shen, Shenggui Li, Hongxin Liu, Yukun Zhou, Tianyi Li, and Yang You. Open-sora: Democratizing efficient video production for all, March 2024. URL https://github.com/hpcaitech/Open-Sora.

---

> > > > > ### Author Response · Authors · 2024-11-22
> > > > > **Response for Reviewer Dx2p**
> > > > >
> > > > > **3. Clarification on experiment and ablation results.**
> > > > >
> > > > > **Q1: The authors fail to convincingly argue why their proposed approach gives a very strong performance compared to previous methods in Section 4.2. What makes the proposed approach very unique to achieve such a performance? Does using multiple kernels give such a performance boost?**
> > > > >
> > > > > **A1:** The performance improvement is attributed to our multi-scale convolution kernels in temporal downsampling module. Compared to other methods [1][2][3], we introduce multi-scale convolution kernels for temporal compression to our UniVAE, which endows it with better capability to capture dynamic patterns of different time-scales in videos. As a result, our UniVAE achieves better reconstruction and generation performance as shown in Tab.1 (main paper) and Tab.2 (main paper).
> > > > >
> > > > > [1] Sijie Zhao, Yong Zhang, Xiaodong Cun, Shaoshu Yang, Muyao Niu, Xiaoyu Li, Wenbo Hu, and Ying Shan. Cv-vae: A compatible video vae for latent generative video models. arXiv preprint arXiv:2405.20279, 2024.
> > > > >
> > > > > [2] Zangwei Zheng, Xiangyu Peng, Tianji Yang, Chenhui Shen, Shenggui Li, Hongxin Liu, Yukun Zhou, Tianyi Li, and Yang You. Open-sora: Democratizing efficient video production for all, March 2024. URL https://github.com/hpcaitech/Open-Sora.
> > > > >
> > > > > [3] Liuhan Chen, Zongjian Li, Bin Lin, Bin Zhu, Qian Wang, Shenghai Yuan, Xing Zhou, Xinghua Cheng, and Li Yuan. Od-vae: An omni-dimensional video compressor for improving latent video diffusion model. arXiv preprint arXiv:2409.01199, 2024.
> > > > >
> > > > > **Q2: Details about experiment in Sec.4.4.**
> > > > >
> > > > > **A2:** (1) Dataset: The 1000 videos used in Sec.4.4 for evaluation are selected from WebVid-10M dataset. (2) Performance: When reconstructing 17-frame videos, all three methods can reconstruct video accurately. That's reason why OS-VAE, OD-VAE, and UniVAE achieve comparable performance. On the other hand, for efficiency, we select a small subset from WebVid-10M for evaluation in Sec.4.4 (as described in (1)), which leads to some differences compared to the results in Tab.1, where the whole WebVid-10M dataset is used.
> > > > >
> > > > > **Q3: The Fig.7 in ablation study.**
> > > > >
> > > > > **A3:** Here, present detailed comparison of the fourth images of Fig.7 (main paper) in Fig.4 (response letter), which are the additional intermediate frames generated by UniVAE-V2 and UniVAE, respectively. As we can see, the UniVAE can better preserve hand detail features than UniVAE-V2.
> > > > >
> > > > >
> > > > >
> > > > > **4. The qualitative analysis is limited.**
> > > > >
> > > > > Thanks for your suggestion. We have added more qualitative results in the updated appendix to provide clearer visual comparison.
> > > > >
> > > > > For video reconstruction, we put qualitative results in *``.zip/Qualitative_Results/Reconstruction/Reconstruction_1.mp4''* and *``Reconstruction_2.mp4''*. For video generation, we put them in *``.zip/Qualitative_Results/Generation/Generation.mp4''*.

---

> > > > > > ### Comment · Reviewer_Dx2p · 2024-11-25
> > > > > >
> > > > > > I appreciate the authors' detailed rebuttal and the additional results provided. After carefully reviewing the response, I find that while some of my questions have been clarified, my main concerns remain largely unaddressed. Below are the issues I would like to highlight:
> > > > > >
> > > > > > -    The performance improvement (+3 dB) claimed in Table 1 is puzzling. The baselines compared predominantly use temporal average pooling, yet the performance gap between the UniVAE with temporal pooling and multi-kernel convolution is marginal (less than 0.2 dB). This suggests the improvement cannot be attributed to multi-kernel convolution as claimed. Could the authors clarify if the results in Table 1 were conducted under fair experimental settings?
> > > > > >
> > > > > > -    It is unclear why introducing multi-kernel convolution in the upsampling module leads to performance degradation. This seems counter-intuitive to the purported benefits of multi-kernel convolution discussed in the paper. Could the authors provide more insights into this observation?
> > > > > >
> > > > > > -    The rebuttal on the interpolation model is not convincing. The interpolation decoder appears dependent on a pre-trained (frozen) VAE, relying on its encoded latent and decoded frames. How does this differ fundamentally from training a separate interpolation model? Additionally, the statement "our refinement decoder can directly reconstruct the video and the generated additional frame $[x_1, x_{new}, x_2]$ in a single pass" is unclear. Doesn't the interpolation decoder still depend on the reconstructed frames? If so, how is it distinct from typical interpolation models?
> > > > > >
> > > > > > -   If the primary focus is on video VAE, why is a causal formulation necessary? Would it not be more straightforward to formulate the problem with N frames instead of N+1? The N+1 approach is generally used when the goal is to jointly encode images and videos, which this work does not appear to do.
> > > > > >
> > > > > > -    Could the authors provide results for evaluating the performance of different video VAEs on higher-resolution videos?
> > > > > >
> > > > > > -    The decision to collect a new dataset raises questions. Why not use existing datasets such as WebVid-10M or Panda-70M, which are commonly used in the field? Additionally, there are no details about the newly collected dataset, making reproducibility a significant concern. Without such details, it is unclear how test set contamination was avoided. This is a critical weakness of the work as it stands.
> > > > > >
> > > > > > -   Why weren’t the video generation performance results updated with sufficient training during the rebuttal period?
> > > > > >
> > > > > > -    The rebuttal about the performance improvement compared to baseline methods is not convincing. Please refer to my first comment which is related to this issue as well.
> > > > > >
> > > > > > -   The details regarding the evaluation dataset are still unclear. Are the authors claiming that the entire WebVid-10M dataset, comprising over 10 million videos, was used for inference in Table 1? How did the authors select the 1000 videos used for the analysis in Sec. 4.4?
> > > > > >
> > > > > >
> > > > > > As my primary concerns with the paper remain unresolved, I will be maintaining my initial score.

---

> > > > > > > ### Author Response · Authors · 2024-11-29
> > > > > > > **Response for Reviewer Dx2p**
> > > > > > >
> > > > > > > **Q1: Clarification on experiment settings.**
> > > > > > >
> > > > > > > **A1:** The comparison results in Tab.1 (main paper) for video reconstruction are all evaluated on the whole WebVid-10M validation set. The input videos are all pre-processed to a length of 25 frames with a resolution 256 x 256, following the same settings as OD-VAE. Please refer to Sec.4.1 and OD-VAE for details.
> > > > > > >
> > > > > > > **Q2: It is unclear why introducing multi-kernel convolution in the upsampling module leads to performance degradation. This seems counter-intuitive to the purported benefits of multi-kernel convolution discussed in the paper. Could the authors provide more insights into this observation?**
> > > > > > >
> > > > > > > **A2:** In downsampling, we introduce adopt multi-scale kernels for temporal compression, which allows the encoder **E** to capture temporal dynamics at multiple time scales, reducing information loss caused by single-scale downsampling and improving the quality of the latent representation.
> > > > > > >
> > > > > > > In upsampling, the decoder **D** aims to reconstruct videos from the latent representation **Z**, which relies primarily on the high-quality representation of **Z** rather than performing additional complex feature modeling. The encoder has already encoded the rich temporal information into the latent representation **Z** using multi-scale convolution kernels. The decoder's task is simply to reconstruct the video based on **Z**, without the need for further complex temporal modeling. It should be noted that the latent representation **Z** that input to the decoder **D** is highly compact with low feature dimension, resulting in significantly lower modeling demands for the decoder **D** compared to the encoder **E**. Therefore, incorporating multi-scale kernels into the decoder is unnecessary.
> > > > > > >
> > > > > > > On the other hand, introducing multi-scale convolution kernels again in the decoder could result in redundant modeling of the already existing multi-scale features. This repretition may introduce unnecessary noise or disrupt the original feature structure, ultimately compromising the quality of video reconstruction. Moreover, it increases the model's complexity, which could further degrade the reconstruction performance.
> > > > > > >
> > > > > > > **Q3: The rebuttal on the interpolation model is not convincing. The interpolation decoder appears dependent on a pre-trained (frozen) VAE, relying on its encoded latent and decoded frames. How does this differ fundamentally from training a separate interpolation model? Additionally, the statement "our refinement decoder can directly reconstruct the video and the generated additional frame in a single pass" is unclear. Doesn't the interpolation decoder still depend on the reconstructed frames? If so, how is it distinct from typical interpolation models?**
> > > > > > >
> > > > > > > **A3:** Separate frame interpolation methods use the raw videos to generate intermediate frames, which typically works on pixel space. They take the existing frames as input and generate the intermediate frames. For example, in "VAE + FILM", the FILM takes the reconstructed videos [$y_1$, $y_2$] by decoder $D_1$ as input, and generate the intermediate frames $y_{new}$.
> > > > > > >
> > > > > > > However, our refinement decoder $D_2$ generate additional intermediate frames based on the latent representation **Z** extracted by the encoder, which works on the latent space. The refinement decoder $D_2$ takes the latent representation **Z** as input, and directly generate the reconstructed frame [$y_1$, $y_2$] and intermediate frames $y_{new}$ with a single forward process. Although we inject the features of $D_1$ into $D_2$ to help $D_2$ generate intermediate frames, $D_2$ doesn't directly use the $D_1$'s output as input to generate intermediate frames, which is different from typical interpolation models.
> > > > > > >
> > > > > > > Frame interpolation techiques typically operate in the pixel space, modeling local temporal dynamics of existing videos to generate intermediate frames. In contrast, our refinement decoder takes the latent representation **Z** as input, with its interpolation results relying on the expressive power of **Z**. It fundamentally decodes global temporal dynamics within **Z** rather than merely interpolating local frame-to-frame relationships.

---

> > > > > > > > ### Author Response · Authors · 2024-11-29
> > > > > > > > **Response for Reviewer Dx2p**
> > > > > > > >
> > > > > > > > **Q4: If the primary focus is on video VAE, why is a causal formulation necessary? Would it not be more straightforward to formulate the problem with N frames instead of N+1? The N+1 approach is generally used when the goal is to jointly encode images and videos, which this work does not appear to do.**
> > > > > > > >
> > > > > > > > **A4:** The causal formulation and N+1 frames are the common settings in video VAE for latent video diffusion models. Please refer to OS-VAE, CV-VAE, and OD-VAE for details.
> > > > > > > >
> > > > > > > > **Q5: Could the authors provide results for evaluating the performance of different video VAEs on higher-resolution videos?**
> > > > > > > >
> > > > > > > > **A5:** Of course. Here, we test OD-VAE and UniVAE with videos that are pre-processed to a length of 65 frames with resolution of 512 x 512. We show the comparison results in Table 1. As we can see, our UniVAE shows better reconstruction than OD-VAE when facing with larger resolution.
> > > > > > > >
> > > > > > > > Table 1: Performance comparison between different VAEs on video reconstruction with 512 x 512 resolution.
> > > > > > > > | Method | PSNR | SSIM | LPIPS |
> > > > > > > > | --- | --- | --- | --- |
> > > > > > > > | OD-VAE | 32.25 | 0.9077 | 0.0769 |
> > > > > > > > | UniVAE | 35.61 | 0.9273 | 0.0542 |
> > > > > > > >
> > > > > > > > **Q6: Why weren’t the video generation performance results updated with sufficient training during the rebuttal period?**
> > > > > > > >
> > > > > > > > **A6:** Here, we retrain the Latte equipped with different VAEs on UCF101 dataset for more training steps and show the new comparison in Tab.2. As we can see, our UniVAE still obtain better performance than CV-VAE and OD-VAE.
> > > > > > > >
> > > > > > > > Table 2: Performance comparison of different VAEs on video generation across UCF101.
> > > > > > > > | Method | FVD |
> > > > > > > > | --- | --- |
> > > > > > > > | Latte + CV-VAE | 419.00 |
> > > > > > > > | Latte + OD-VAE | 423.28 |
> > > > > > > > | Latte + UniVAE | 414.22 |
> > > > > > > >
> > > > > > > > **Q7: The usage of WebVid-10M dataset.**
> > > > > > > >
> > > > > > > > **A7:** In lines 363-364 in main paper, we have claimed that we use the validation set of WebVid-10M to evaluate the video reconstruction performance, following the setting in OD-VAE. The 1000 videos in Sec.4.4 are randomly selected from WebVid-10M validation set, as claimed in lines 425-426.

---

> > > > > > > > ### Comment · Reviewer_Dx2p · 2024-12-02
> > > > > > > >
> > > > > > > > Thank you for your response.
> > > > > > > >
> > > > > > > > Unfortunately, the authors have not directly addressed several of my concerns. For example, they have not explained why the use of multi-kernel convolution results in only a marginal improvement compared to temporal average pooling within their model, while outperforming other baselines—also employing temporal average pooling—by more than 3 dB on the PSNR metric. Additionally, the authors did not clearly answer whether the experiments were conducted under fair settings. Furthermore, questions regarding the training dataset remain unanswered.
> > > > > > > >
> > > > > > > > As a result, I will maintain my current score.

---

> > > > > > > > > ### Author Response · Authors · 2024-12-03
> > > > > > > > > **Response for Reviewer Dx2p**
> > > > > > > > >
> > > > > > > > > Thank you for your suggestions.
> > > > > > > > >
> > > > > > > > > The training dataset contains about 1.5M videos collected from the Internet. Since the video quality of WebVid-10M and Panda-70M is limited, we use our private data for training VAE. In fact, for better performance, most VAEs don't adopt WebVid-10M or Panda-70M for training. For fair comparison, we have also trained the pre-trained OD-VAE with our private data, and then compare it with pre-trained OD-VAE. The results show that OD-VAE trained with our private data has comparable and even a little worse performance than pre-trained OD-VAE, which proves that the training data have little impact on the final performance. As for pooling, we need more experiments for deeper analysis. However, it is not the focus of our paper.

---

### Author Response · Authors · 2024-11-22
**We sincerely thank all the reviewers for their careful reading and constructive comments.**

We sincerely thank all the reviewers for their careful reading and constructive comments, which have been invaluable in improving our work. In response to the reviewer's comments, **we have uploaded our supplementary materials,** which include the **complete responses** (at **.zip/UniVAE_Rebuttal_Response_Letter.pdf**) along with the relevant figures and tables. **We sincerely invite the reviewers to refer to these materials for a better reading experience**. We hope that our response satisfactorily addresses your concerns.

---

### Meta-Review · Area_Chair_56XJ · 2024-12-18

**Metareview:**

Summary: Proposes two techniques to a well studied Video VAE architecture. 1) a multi-scale temporal downsampling via 3D convolutions of varying temporal kernel size of 3, 5, 7, to 9. 2) an interpolation decoder that accepts the VAE encoder latents and outputs upsampled video frames. The model is trained in a two-stage manner, with the first being for the basic VAE, and the second for the interpolation decoder.
Strength: The paper is clear. The idea presented is straightforward and easy to understand. Some ablation experiments are well thought out and executed, like the ablation on effectiveness on long video, and to some degree the ablation on the effect of multi-scale temporal downsampling.

Weakness: The problem definition and choice of design is a little unmotivated. Why is causality necessary in the method design? Its role is not clear and results for images are missing. Why is multi-scale only relevant in downsampling? If larger context is really necessary, have you considered studying the role of temporal attention, which can capture global temporal context and is commonly employed in this domain? There is also limited novelty. Joint spatio-temporal modeling is not new in video VAE research (see CogVideoX by Yang et.al). The benefit the interpolation decoder is not clear and misses critical comparison with latent-interpolators (see LDMVFI by Danier et al). Finally, I looked at the supplementary results and the reconstruction video results are rather weak. The presented examples are rather simple global camera movements, and even in this case, flickering and low image quality can be seen obviously.

Rejection Reason: See weakness.

**Additional Comments On Reviewer Discussion:**

The paper received 2x rejection and 2x marginally above acceptance threshold. Reviewers raised a number of points. There is a healthy discussion between the authors and reviewers. Some concerns were addressed, and some were missing, like the impact of multi-scale temporal downsampling, and the role of the interpolation decoder.

---

### Decision · Program_Chairs · 2025-01-22

Reject